# Less Is More: Elevating RAG via Performance-Driven Context Compression

**Ziqiang Cui** [1]  **Yunpeng Weng** [2]  **Xing Tang** [3]  **Peiyang Liu** [4]  **Shiwei Li** [2]  **Bowei He** [5]  **Jiamin Chen** [1]
**Yansen Zhang** [1]  **Xiuqiang He** [3]  **Rui Zhang** [2]  **Chen Ma** [1]

## Abstract

Retrieval-Augmented Generation (RAG) has emerged as a promising paradigm for improving the timeliness of knowledge updates and the factual accuracy of large language models. However, incorporating a large volume of retrieved documents significantly increases input length, leading to prohibitive computational costs. Existing compression approaches often compromise task performance, primarily due to their reliance on predefined heuristics. These heuristics fail to ensure that the compressed context is conducive to the generation tasks. To address these limitations, we propose CORE-RAG, a novel framework for context compression in RAG systems. CORE eliminates reliance on proxy heuristics through a performance-driven learning framework, which directy utilizes task performance as a feedback signal to iteratively refine the compressor policy. Prior to this optimization process, we incorporate a knowledge distillation phase to initialize the compressor with a robust policy. Extensive experiments demonstrate the superiority of our approach. At a high compression ratio of 3%, CORE not only avoids performance degradation but also improves the average Exact Match (EM) score by 3.3 points compared to using full documents. Our code is available at https://github.com/ziqiangcui/CORE-RAG-ICML26.

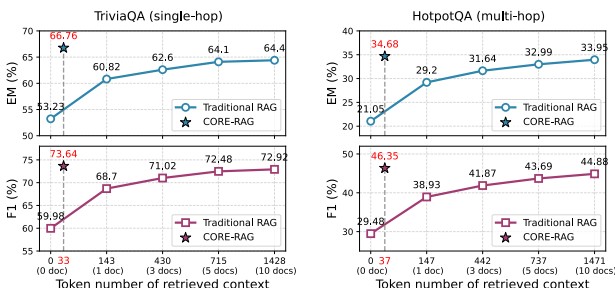

*Figure 1.* Performance evolution with an increasing number of retrieved documents on two datasets. Traditional RAG requires more documents for better performance, while our method CORE-RAG achieves superior results with significant context compression.

## 1. Introduction

Large Language Models (LLMs) have advanced rapidly in recent years, demonstrating significant performance improvements across diverse tasks due to their emergent capabilities in semantic understanding and reasoning. Despite these advancements, LLMs continue to struggle with knowledge updates and the generation of factually accurate responses (Fan et al., 2024). To mitigate these limitations, Retrieval-Augmented Generation (RAG) has emerged as a robust solution. By retrieving the most relevant documents from external knowledge bases and prepending them as contextual information to the original input, RAG substantially enhances the performance of LLMs on knowledge-intensive Question Answering (QA) tasks (Ram et al., 2023).

While RAG offers demonstrable benefits, recent research indicates that its performance is closely tied to context length (Liu et al., 2023). To investigate this relationship, we conducted an experiment on QA tasks where a retriever identified the top-$k$ most relevant documents for a given question. These documents were prepended to the original question to form the prompt used by the LLM to generate an answer. We analyzed how the answer accuracy evolved as the number of retrieved documents increased. As illustrated in Figure 1, starting from the lowest accuracy in the absence of retrieved documents (the non-RAG baseline), performance consistently improved with the addition of context, eventually surpassing the baseline by over 10 Exact Match (EM) points when 10 documents were provided. However, these

---

[1]City University of Hong Kong, Hong Kong SAR, China [2]Huazhong University of Science and Technology, Wuhan, China [3]Shenzhen Technology University, Shenzhen, China [4]Peking University, Beijing, China [5]Mohamed bin Zayed University of Artificial Intelligence, Abu Dhabi, UAE. Correspondence to: Xing Tang <xing.tang@hotmail.com>, Xiuqiang He <hexiuqiang@sztu.edu.cn>, Chen Ma <chenma@cityu.edu.hk>.

*Proceedings of the 43rd International Conference on Machine Learning*, Seoul, South Korea. PMLR 306, 2026. Copyright 2026 by the author(s).

performance gains are accompanied by two significant limitations. First, processing larger contexts creates significant computational overhead (Xu et al., 2024). As shown in the figure, increasing documents from 0 to 10 surges the encoding tokens from 0 to 1428. Second, LLMs struggle to leverage long contexts effectively, often suffering from the "lost-in-the-middle" phenomenon (Liu et al., 2023).

These limitations have motivated recent research into compressing the retrieved context (Wu et al., 2025; Jin et al., 2024a; Zhang et al., 2024a). Prominent approaches include document summarization (Xu et al., 2024), key information extraction (Cao et al., 2024), evidence construction (Jin et al., 2024b), and information-theoretic noise filtering (Zhu et al., 2024). Despite recent progress, these methods exhibit several notable shortcomings. First, these methods all entail a performance compromise, rendering them unsuitable for accuracy-sensitive applications. Second, most methods rely on proxy heuristics designed to guide the compression. These strategies include maximizing mutual information between the source text and summary, mimicking a teacher model's output, or selecting content based on lexical overlap metrics like BM25 (Robertson et al., 1995). While effective for general summarization, these learning objectives fail to consider task performance. This deficiency arises because there is no perfect ground-truth signal to indicate what constitutes an optimal summary for the task performance. Third, some compression models (Zhu et al., 2024) are comparable in size to the downstream LLMs they serve. This incurs substantial computational overhead, negating the efficiency benefits intended by the compression.

In light of these shortcomings, we propose **CORE-RAG**, a lightweight compression framework for RAG. Unlike prior approaches, CORE explicitly aligns the compression strategy with task performance, effectively eliminating performance trade-offs. Given the impracticality of obtaining "gold" summary labels, we formulate compression as a decision-making process where the compressor functions as a policy. This policy is optimized end-to-end via group relative comparisons (Shao et al., 2024), utilizing task performance as direct feedback. To ensure a robust initial policy, we introduce a preliminary warm-up phase where the compressor is initialized via knowledge distillation. Furthermore, our compressor is designed to be significantly smaller than the downstream LLM, which substantially reduces the computational overhead associated with encoding retrieved documents. Remarkably, CORE achieves superior performance with drastically reduced context. As illustrated in Figure 1, CORE (star markers) outperforms the 10-document full-context baseline while utilizing only 3% of the tokens.

Our contributions are summarized as follows:

- We introduce a novel context compression framework for RAG. Unlike prior methods, our approach directly optimizes the context compressor in a performance-driven manner, eliminating reliance on proxy heuristics.
- We introduce a knowledge distillation phase that provides a robust initialization and enhances training stability for performance-driven learning, thereby unlocking its full potential.
- Extensive experiments demonstrate that CORE not only outperforms existing compression baselines but also achieves higher accuracy than standard full-context RAG while significantly reducing context length, truly embodying the "less is more" principle.

**Conflict of Interest Disclosure** The authors declare that they have no conflicts of interest.

## 2. Related Work

### 2.1. Context Compression for RAG

RAG enhances the performance of LLMs on knowledge-intensive tasks by retrieving relevant documents and prepending them as contextual information (Ram et al., 2023; Fan et al., 2024). However, this paradigm requires the LLM to process significantly longer token sequences, resulting in substantially increased computational costs. Researchers have begun to explore methods for compressing context in RAG systems (Rau et al., 2024; Wu et al., 2025; Louis et al., 2025; Jin et al., 2024a; Li et al., 2024a;b; Zhang et al., 2024a). Some soft compression techniques encode contextual inputs into dense embeddings (Chevalier et al., 2023). A representative example is xRAG (Cheng et al., 2024), which employs Multi-layer Perceptrons to map context into fixed-size latent vectors. Nevertheless, such soft approaches often lack interpretability and transferability, and degrade task performance. In addition, some hard methods paraphrase input context through summarization or selectively retain tokens to reduce context length. For instance, Xu et al. (2024) propose compressing retrieved documents into textual summaries, training the compressor through data selection and distillation. In parallel, both LongLLM-Lingua (Jiang et al., 2024) and QGC (Cao et al., 2024) leverage the input question to guide the compression process, with LongLLMLingua further incorporating document reordering to mitigate position bias. Jin et al. (2024b) refine retrieved documents into key supporting evidence. Meanwhile, Kim & Thorne (2025) train a compressor to extract critical information using reward functions based on predefined heuristic rules. Additionally, Zhu et al. (2024) present an information-theoretic approach called NoiseFilter-IB, which filters noise by maximizing the mutual information between the compressed context and the ground-truth output. Beyond RAG-specific techniques, general task-agnostic prompt compression methods have also been widely ex-

plored to reduce computational overhead. For example, DAC (Zhao et al., 2025) proposes a dynamic attention-aware approach that integrates information entropy and attention scores to iteratively prune less informative tokens without relying on specific downstream task clues. However, due to the absence of ground-truth compression labels, most of these methods are heuristic in nature; consequently, they typically incur a performance penalty on downstream tasks. In contrast, our method adopts a performance-driven optimization approach to address these limitations.

## 2.2. Application of Reinforcement Learning

Reinforcement learning has recently achieved notable success (Liu et al., 2024; Shao et al., 2024; Guo et al., 2025; Li & Ramakrishnan, 2025). Building on these advances, several studies have applied RL to improve RAG (Ke et al., 2024). For example, Kulkarni et al. (2024) use RL to autonomously decide whether to retrieve documents, while Zhang et al. (2024b) employ RL to optimize the ranking of retrieved documents. Similarly, MMOA-RAG (Chen et al., 2025b) enhances RAG through multi-agent RL. Beyond RAG, RL has been applied in other domains; for instance, TACO (Shandilya et al., 2025) utilizes RL to compress prompts through token-level keep-or-drop decisions. However, such prompt compression methods are ill-suited for RAG, and our approach differs fundamentally in both nature and performance. First, we employ a generative compressor capable of rephrasing and synthesizing content, rather than relying on binary token-level actions. Most importantly, we optimize compression using a direct performance-driven reward. This enables us to achieve true lossless compression, standing in stark contrast to the performance degradation observed in TACO (Shandilya et al., 2025).

Moreover, a line of research has utilized RL to integrate search with reasoning in a step-by-step manner (Singh et al., 2025). For instance, Chen et al. (2025a) introduce a framework called ReSearch, which trains LLMs to reason with search using RL. Related approaches include R1-Searcher (Song et al., 2025), WebThinker (Li et al., 2025), and Deep-Researcher (Zheng et al., 2025). However, these methods differ fundamentally from our problem setting, precluding a fair comparison. Typically, such approaches involve directly training the LLM generator—often a large-scale model with a high parameter count. However, this strategy becomes infeasible when the model is a black box—the precise setting addressed in this paper where internal weights are inaccessible. Furthermore, these methods often introduce extensive internal reasoning processes that substantially increase context length and inference latency. In contrast, our approach treats the generator LLM as a fixed black-box model, training only a lightweight plug-in compressor to produce document summaries. This design significantly improves both training and inference efficiency.

## 3. Methodology

This section presents our proposed framework, Performance-Driven **CO**mpression via **RE**inforcement learning for **RAG** (**CORE-RAG**), as illustrated in Figure 2. We begin by formally defining the context compression task within the RAG paradigm in Section 3.1. Section 3.2 details our training methodology, followed by a description of the inference pipeline and deployment strategy in Section 3.3.

### 3.1. Problem Formulation

We adopt the same problem formulation as RECOMP (Xu et al., 2024). Given an input question $q$, a target output $y$, and a set of $k$ retrieved documents $D$, our objective is to compress $D$ with respect to $q$ into a summary $s$ that preserves the most useful information while using significantly fewer tokens. This summary $s$ is then prepended to the original input $q$ and fed into a blackbox LLM to generate the final response for the downstream task. The process involves two key components: a compressor $\pi_\theta \colon (q, D) \mapsto s$ and a blackbox large language model $M \colon (s, q) \mapsto \hat{y}$, which generates the predicted answer $\hat{y}$. Note that we treat $M$ as a black-box system and focus exclusively on training the compressor $\pi_\theta$. The compressor is intentionally designed to be significantly smaller than $M$ to reduce computational cost.

### 3.2. Performance-Aware Training Framework

The compressor aims to generate compressed documents tailored to maximize utility for the target LLM ($M$) on downstream tasks. Achieving this is challenging due to the absence of direct supervision. To address this, we formulate compression as a decision-making process and employ reinforcement learning. An overview of our training framework is depicted in the lower portion of Figure 2, with detailed procedures provided in the subsequent subsections.

#### 3.2.1. DISTILLATION FOR WARM-START

Given that our compression model is intentionally designed with significantly fewer parameters than the downstream LLM, its inherent capacity for query-focused context compression is limited. To address this limitation and provide a robust initial policy for RL, we employ knowledge distillation to initialize the compression model. Specifically, we utilize a large-scale language model (DeepSeek-V3) as a teacher to generate summaries of the retrieved documents $D$ associated with each question $q$ in the training dataset $\mathcal{X}$. The prompt template used for summary generation is illustrated in the upper portion of Figure 3.

To ensure the construction of high-quality training data, we evaluate the performance of the downstream LLM ($M$) on the QA task under two distinct conditions: (1) with the teacher-generated summary $\hat{s}$ prepended to the input

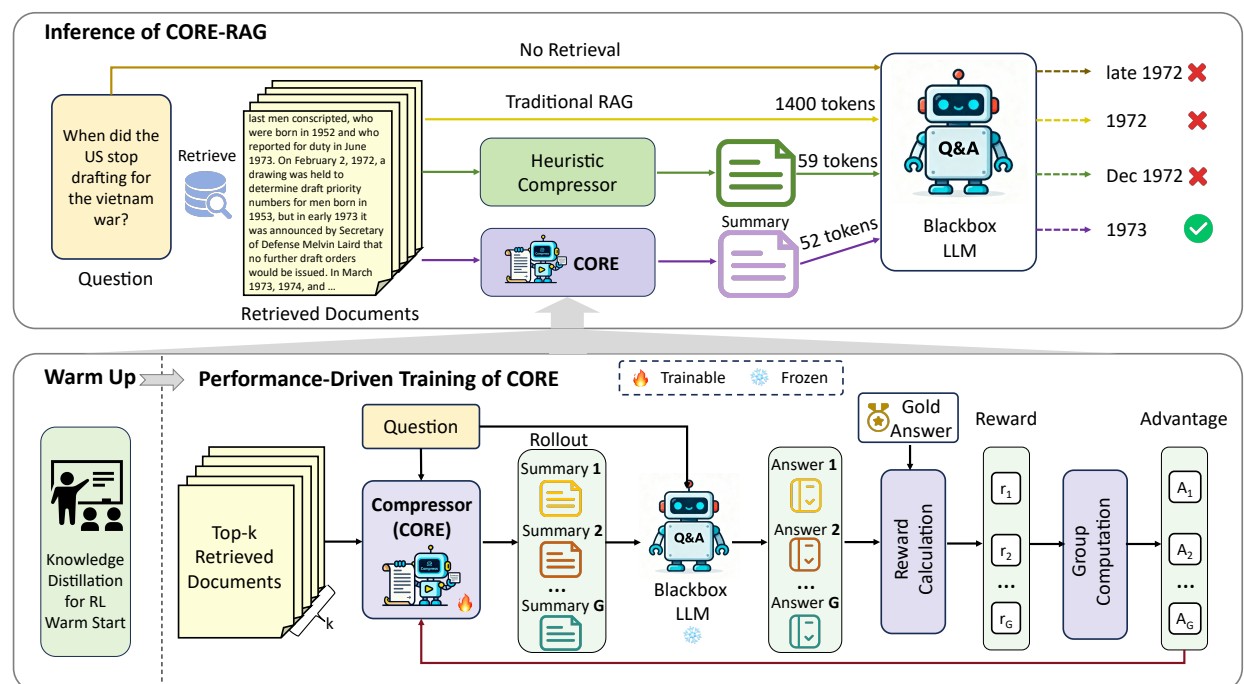

*Figure 2.* Overview of our method CORE. The upper section illustrates the inference pipeline. The lower section depicts the training framework for the compression model.

question $q$, and (2) with the original question $q$ alone as the prompt. Let $p_{summary}$ and $p_{original}$ denote the corresponding performance scores. We then curate the training data based on the following criteria:

- We retain instances where $p_{summary} > p_{original}$, indicating that the summary effectively enhances model performance. For these samples, the teacher-generated summary is designated as the target output $\hat{s}$.
- For instances where the model answers correctly without the summary ($p_{original} = 1$) but performance degrades when the summary is included ($p_{summary} < p_{original}$), we set the target summary $\hat{s}$ to an empty string. This strategy trains the model to avoid generating detrimental summaries or unnecessary noise.

All other instances are discarded. The resulting dataset, denoted as $\mathcal{X}_f$, is used for supervised fine-tuning. The training prompt employed is identical to that of the teacher model, as illustrated in the upper portion of Figure 3. Formally, the fine-tuning objective is defined as:

$$\mathcal{L}_d = -\frac{1}{|\mathcal{X}_f|} \sum_{(q,D,\hat{s}) \in \mathcal{X}_f} \sum_{t=1}^{|\hat{s}|} \log P_{\pi_\theta}(\hat{s}_t \mid q, D, \hat{s}_{<t}), \quad (1)$$

where $\hat{s}_t$ is the $t$-th token of the target summary $\hat{s}$, and $\hat{s}_{<t}$ represents the preceding tokens.

### 3.2.2. PERFORMANCE-DRIVEN OPTIMIZATION

Subsequent to the distillation phase, the compressor acquires a preliminary capacity for summarization. However, because summaries generated by even the largest teacher models are not guaranteed to be optimal for downstream task performance, further performance-driven optimization is required. We formulate this optimization as a reinforcement learning problem, where the compressor $\pi_\theta$ operates as a policy, generating a summary of the retrieved documents based on the input question. The prompt for $\pi_\theta$ is identical to the one used in the distillation phase, as illustrated in the upper portion of Figure 3. The quality of the summary is then evaluated by a reward function that directly reflects the task performance. The ultimate objective is to optimize the compressor's parameters to maximize the expected cumulative reward, thereby aligning its outputs directly with the task performance.

Specifically, we employ the Group Relative Policy Optimization (GRPO) algorithm (Shao et al., 2024). Unlike Proximal Policy Optimization (PPO), which necessitates a separate critic model, GRPO estimates the baseline directly from a group of rollouts. To simplify notation, we define the input state as $x = (q, D)$, representing the pair of the input question and the retrieved documents. Given an existing policy $\pi_{\theta_{old}}$ and a reference policy $\pi_{\theta_{ref}}$, the GRPO objective optimizes the compressor policy $\pi_\theta$ by sampling a group of $G$ summaries $\{s_i\}_{i=1}^{G} \sim \pi_{\theta_{old}}(\cdot|x)$ for each input

---

**Prompt Template for Compression**

Compress the information in the retrieved documents into a 2-sentence summary that could be used to answer the question. If the documents do not contain relevant information, simply output " ".

Question: {*Question*}
Retrieved documents: {*Documents*}
Compressed documents:

**Prompt Template for Answer Generation**

Answer the question. Respond ONLY with the exact answer in the same format as the examples. Do NOT add any extra text, explanations, or punctuation. Do NOT include "Answer:" or any similar prefix in your response.

{*Examples*}

{***Compressed documents***}
Question: {*Question*}
Answer:

---

*Figure 3.* Prompt template.

$x$ drawn from the training dataset $\mathcal{X}$. The objective function is formulated as:

$$\mathcal{J}(\theta) = \mathbb{E}_{x \sim \mathcal{X}, \ \{s_i\}_{i=1}^G \sim \pi_{\theta_{\text{old}}}(\cdot|x)}$$

$$\frac{1}{G} \sum_{i=1}^G \left[ \min\left( \rho_i(\theta) A_i, \ \text{clip}\left(\rho_i(\theta), 1-\epsilon, 1+\epsilon\right) A_i \right) \right.$$

$$\left. - \beta \mathbb{D}_{\text{KL}}\left( \pi_\theta \ \| \ \pi_{\theta_{\text{ref}}} \right) \right], \tag{2}$$

where $\rho_i(\theta) = \frac{\pi_\theta(s_i|x)}{\pi_{\theta_{\text{old}}}(s_i|x)}$ denotes the probability ratio between the current policy and the old policy. $A_i = (r_i - \text{mean}(\{r_j\}_{j=1}^G))/\text{std}(\{r_j\}_{j=1}^G)$ is the normalized advantage of the $i$-th rollout within the group, $\epsilon$ is the clipping ratio, and $\beta$ is the coefficient for the KL divergence penalty.

### 3.2.3. REWARD DESIGN

To effectively guide the compressor towards generating utility-preserving summaries, we design a reward function that directly reflects end-task performance.

**Answer Generation.** Crucially, the reward is not derived directly from the compressor's summary itself. Instead, the generated summary $s$ is prepended to the original question $q$ to form a joint input for the LLM ($M$). The LLM then generates a predicted answer: $\hat{y} = M(s, q)$. The prompt used for this process is illustrated in the bottom half of Figure 3. The reward is computed by comparing the predicted answer $\hat{y}$ against the ground truth answer $y$. Note that throughout the training phase, the LLM ($M$) remains frozen and serves solely as an answer generator. This characteristic distinguishes our work from many other RL-based approaches.

**Reward Formulation.** We employ a composite reward function consisting of two rule-based metrics:

- **EM Reward** ($r_{\text{EM}}$). We utilize EM as the primary metric. This binary reward assigns a value of 1 solely when the

prediction is identical to the ground truth:

$$r_{\text{EM}} = \mathbb{I}(\hat{y} = y) = \begin{cases} 1 & \text{if } \hat{y} = y, \\ 0 & \text{otherwise.} \end{cases} \tag{3}$$

- **F1 Reward** ($r_{\text{F1}}$). Relying exclusively on EM can lead to sparse reward signals. To provide finer-grained supervision for partially correct answers, we incorporate the token-level F1 score:

$$r_{\text{F1}} = \frac{2 \times I_N}{P_N + R_N}, \tag{4}$$

where $P_N$ and $R_N$ denote the token counts in the predicted answer $\hat{y}$ and the gold answer $y$, respectively, and $I_N$ represents the count of overlapping tokens.

The final reward is a weighted sum of these components:

$$r = r_{\text{EM}} + \alpha \cdot r_{\text{F1}}, \tag{5}$$

where $\alpha \in (0, 1]$ is a hyperparameter balancing the contribution of the partial-match signal.

### 3.3. Inference and Deployment

During the inference phase, the trained compressor $\pi_\theta$ operates as a lightweight, standalone pre-processing module. Specifically, given an input question $q$ and a set of retrieved documents $D$, the compressor generates a concise summary $s = \pi_\theta(q, D)$ using greedy decoding. This generated summary is then prepended to the original question $q$ to form the final input context $(s, q)$ for the downstream LLM $M$. The downstream model then generates the final response $\hat{y} = M(s, q)$. By substituting the voluminous raw documents $D$ with a compact summary $s$, CORE significantly reduces the LLM's input sequence length, thereby lowering inference latency and computational costs. We provide an analysis of the efficiency of our method in Appendix B.

### 3.4. Discussion on Open-ended Tasks

For open-ended settings, the reward should be replaced with evaluation criteria appropriate for generative tasks. For long-form generation, rewards should focus on instruction-following and coherence, scored via an LLM-as-a-Judge or preference reward models. For summarization, the reward should balance information coverage and factual consistency using metrics like BARTScore or ROUGE, together with hallucination penalties. For dialogue, the reward can be turn-level coherence and persona consistency. Importantly, only the reward changes; the framework itself remains the same: optimizing a lightweight compressor for downstream task performance. Therefore, CORE should generalize naturally beyond extractive QA to open-ended settings with appropriate rewards.

# 4. Experiments

## 4.1. Experimental Settings

### 4.1.1. DATASETS AND EVALUATION METRICS.

We evaluate our method on four benchmark datasets: two single-hop question-answering datasets, Natural Questions (NQ) (Kwiatkowski et al., 2019) and TriviaQA (Joshi et al., 2017), as well as two multi-hop question-answering datasets, HotpotQA (Yang et al., 2018) and 2WikiMultihopQA (Ho et al., 2020). Results are reported on the test sets of NQ and TriviaQA, as well as the development sets of HotpotQA and 2WikiMultihopQA. Following RECOMP (Xu et al., 2024), the performance is measured using Exact Match and token-level F1 scores, while efficiency is assessed by the number of tokens provided in the context.

### 4.1.2. COMPRESSION MODEL ($\pi_\theta$) AND LLM ($M$).

We trained our compression model $\pi_\theta$ using Qwen2.5-1.5B-Instruct to generate summaries; the results reported in Table 1 are based on this configuration. To assess the robustness of our approach across different models, we also trained compressors using Llama3.2-1B-Instruct and Llama3.2-3B-Instruct, as detailed in Section 4.5.

We use Qwen2.5-14B-Instruct as the LLM $M$ to generate predicted answers. To assess the generalization capability of our method, we also evaluate its performance when transferred to a different LLM, Llama-3.1-8B-Instruct.

### 4.1.3. RETRIEVAL CORPUS AND RETRIEVERS.

Following previous studies (Xu et al., 2024), we use the Wikipedia corpus from December 20, 2018, as the retrieval source. The articles are segmented into non-overlapping 100-word documents. To ensure that our method is not dependent on a specific retriever, we experiment with several mainstream retrievers. Specifically, we use DPR (Karpukhin et al., 2020) for NQ, a hybrid of DPR and BM25 (Robertson et al., 1995) for TriviaQA, and the Contriever model (Izacard et al., 2021) trained on the MS MARCO dataset (Nguyen et al., 2016) for HotpotQA and 2WikiMultihopQA.

### 4.1.4. BASELINES AND IMPLEMENTATION DETAILS.

To evaluate the effectiveness of our method, we conduct a comprehensive comparison against a diverse set of baselines. First, we evaluate the **full-context approach**, which adheres to the standard RAG setup by prepending the top-$k$ retrieved documents (for $k \in \{1, 3, 5, 10\}$) directly to the prompt. We also assess the traditional **BM25** algorithm, which selects sentences based on lexical similarity to the input. Additionally, we compare our method against off-the-shelf LLMs: **Qwen2.5-1.5B-Instruct**, which shares the same parameter scale as our approach, and **DeepSeek-V3**,

a 671B-parameter model that serves as a large-scale performance reference. Furthermore, we benchmark against state-of-the-art context compression methods to ensure a comprehensive evaluation. These include **RECOMP** (Xu et al., 2024), where we evaluate both its abstractive and extractive variants; **NoiseFilter-IB** (Zhu et al., 2024), an information-theoretic approach designed to filter irrelevant noise; **LongLLMLingua** (Jiang et al., 2024), which specializes in compressing long-context prompts for LLMs; and **QGC** (Cao et al., 2024), a query-guided compression technique. To ensure a fair comparison, all trainable baselines utilize the same model for initialization. Implementation details of our method are provided in Appendix C.

## 4.2. Overall Performance

The detailed comparison results are presented in Table 1. For fair comparison, all trainable methods were trained using the same backbone model, Qwen2.5-1.5B-Instruct, with 5 documents used for compression training. From the results, we can observe several key findings:

- **CORE achieves higher accuracy than traditional full-context RAG with significant context compression.** As shown in Table 1, our method reduces the input size to approximately 6% of the full-context RAG. Crucially, this significant reduction incurs no loss in downstream performance. In fact, our approach improves EM scores by 1.5 to 6.9 points on all four datasets compared to the full-context baseline.

- **CORE outperforms all compression baselines.** All baseline compression methods result in varying degrees of performance degradation compared to the full-context baseline. Specifically, BM25 and the off-the-shelf Qwen2.5-1.5B-Instruct model cause substantial performance drops. Surprisingly, even DeepSeek-V3 (671B parameters) fails to match the full-context baseline on the NQ and 2Wiki datasets. Similarly, trainable baselines (RECOMP, NoiseFilter-IB, LongLLMLingua, and QGC) exhibit performance declines generally ranging from 2 to 6 EM points across all datasets relative to the no-compression setting. In contrast, our method, CORE, achieves superior performance. It not only surpasses comparable compression methods by 4–5 EM points but also outperforms the exponentially larger DeepSeek model, clearly demonstrating the efficacy of our performance-driven optimization.

- **CORE demonstrates robust generalization to longer contexts.** Although our compressors were trained solely on 5-document inputs, we apply them directly to the top-10 document setting without any retraining to evaluate their capability for length generalization. The results in the bottom section of Table 1 show that CORE successfully extrapolates to this longer context. It continues to

*Table 1.* QA results on four benchmarks with Qwen2.5-14B-Instruct as the LLM ($M$). Both our compressor (CORE) and all trainable compression baselines (RECOMP, NoiseFilter-IB, LongLLMLingua, QGC) are trained using Qwen2.5-1.5B-Instruct to ensure a fair comparison. The reported token counts represent the length of in-context documents, excluding few-shot examples.

| | NQ | | | TriviaQA | | | HotpotQA | | | 2WikiMultihopQA | | |
|---|---|---|---|---|---|---|---|---|---|---|---|---|
| | EM | F1 | # tok | EM | F1 | # tok | EM | F1 | # tok | EM | F1 | # tok |
| No Retrieval | 21.36 | 30.97 | 0 | 53.23 | 59.98 | 0 | 21.05 | 29.48 | 0 | 26.11 | 29.51 | 0 |
| *RAG without compression (full-context)* | | | | | | | | | | | | |
| Top 1 Document | 34.46 | 44.41 | 142 | 60.82 | 68.70 | 143 | 29.20 | 38.93 | 147 | 26.79 | 31.87 | 153 |
| Top 3 Documents | 37.78 | 48.45 | 427 | 62.60 | 71.02 | 430 | 31.64 | 41.87 | 442 | 27.89 | 33.58 | 460 |
| Top 5 Documents | 38.03 | 49.16 | 712 | 64.10 | 72.48 | 715 | 32.99 | 43.69 | 737 | 29.64 | 35.21 | 766 |
| Top 10 Documents | 38.67 | 50.03 | 1425 | 64.40 | 72.92 | 1428 | 33.95 | 44.88 | 1471 | 31.04 | 36.75 | 1531 |
| *Compression of top 5 documents* | | | | | | | | | | | | |
| BM25 | 25.23 | 36.47 | 37 | 55.36 | 63.90 | 39 | 24.18 | 35.73 | 71 | 25.42 | 30.29 | 68 |
| Qwen2.5-1.5B-Instruct | 31.94 | 43.03 | 36 | 57.99 | 66.70 | 30 | 27.36 | 37.47 | 33 | 25.93 | 31.18 | 32 |
| DeepSeek-V3 (671B) | 37.73 | 50.39 | 54 | 64.13 | 73.20 | 50 | 33.59 | 44.83 | 48 | 27.99 | 32.67 | 92 |
| RECOMP-Abs (1.5B) | 34.18 | 46.26 | 58 | 60.31 | 68.50 | 53 | 28.96 | 39.95 | 56 | 30.25 | 36.73 | 52 |
| RECOMP-Ext (1.5B) | 33.84 | 46.05 | 56 | 60.18 | 68.39 | 48 | 29.93 | 41.09 | 45 | 30.78 | 37.07 | 51 |
| NoiseFilter-IB (1.5B) | 35.15 | 45.94 | 48 | 59.51 | 68.15 | 35 | 27.97 | 38.62 | 38 | 27.85 | 34.69 | 40 |
| LongLLMLingua (1.5B) | 33.65 | 43.15 | 152 | 58.96 | 66.82 | 148 | 28.03 | 38.49 | 149 | 29.37 | 33.62 | 153 |
| QGC (1.5B) | 36.23 | 45.88 | 49 | 61.02 | 68.45 | 47 | 29.16 | 40.05 | 45 | 31.14 | 36.83 | 51 |
| **CORE (1.5B)** | **41.02** | **50.40** | **46** | **65.63** | **72.55** | **32** | **33.67** | **45.06** | **36** | **36.72** | **42.05** | **49** |
| *Generalization to top 10 documents (with the compressor trained on top 5 documents)* | | | | | | | | | | | | |
| BM25 | 25.91 | 36.88 | 38 | 55.28 | 63.16 | 37 | 23.49 | 35.01 | 68 | 25.61 | 30.54 | 65 |
| Qwen2.5-1.5B-Instruct | 32.94 | 44.84 | 40 | 58.45 | 67.31 | 33 | 28.17 | 38.48 | 36 | 26.22 | 31.57 | 34 |
| DeepSeek-V3 (671B) | 37.79 | 51.07 | 56 | 65.29 | 74.45 | 53 | 34.62 | 45.69 | 50 | 29.00 | 34.64 | 40 |
| RECOMP-Abs (1.5B) | 34.40 | 46.93 | 59 | 61.42 | 69.88 | 52 | 31.54 | 42.92 | 52 | 31.98 | 38.16 | 49 |
| RECOMP-Ext (1.5B) | 33.96 | 46.34 | 60 | 61.03 | 69.51 | 50 | 31.92 | 43.18 | 55 | 32.52 | 38.87 | 44 |
| NoiseFilter-IB (1.5B) | 35.36 | 46.24 | 50 | 59.92 | 68.32 | 38 | 28.21 | 38.83 | 38 | 28.63 | 35.16 | 42 |
| LongLLMLingua (1.5B) | 33.78 | 43.37 | 154 | 59.17 | 66.97 | 150 | 28.33 | 38.95 | 148 | 29.62 | 34.11 | 151 |
| QGC (1.5B) | 36.03 | 45.62 | 50 | 61.23 | 68.74 | 49 | 29.12 | 39.63 | 46 | 31.71 | 37.52 | 50 |
| **CORE (1.5B)** | **41.88** | **51.26** | **52** | **66.76** | **73.64** | **33** | **34.68** | **46.35** | **37** | **37.99** | **43.28** | **48** |

*Table 2.* Ablation study on all datasets.

| Dataset | Metric | w/o distillation | w/o RL | CORE |
|---|---|---|---|---|
| NQ | EM | 36.37 | 34.18 | **41.02** |
| | F1 | 46.91 | 46.26 | **50.40** |
| TQA | EM | 65.23 | 60.31 | **65.63** |
| | F1 | 72.41 | 68.50 | **72.55** |
| HotpotQA | EM | 32.01 | 28.96 | **33.67** |
| | F1 | 42.73 | 39.95 | **45.06** |
| 2Wiki | EM | 31.40 | 30.25 | **36.72** |
| | F1 | 36.89 | 36.73 | **42.05** |

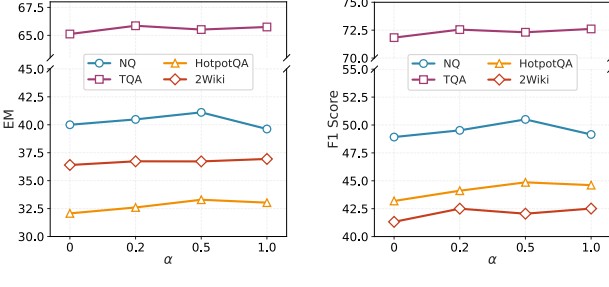

*Figure 4.* The impact of $\alpha$.

### 4.3. Ablation Study

Table 2 presents an ablation study on the two training stages of our method. Specifically, "w/o distillation" refers to training the compressor directly with GRPO, bypassing the warm-start phase, while "w/o RL" denotes relying solely on the distillation stage. The results show that removing either stage leads to performance degradation, confirming the necessity of both. Notably, the performance drop is more

outperform the uncompressed 10-document baseline and ranks best among all compression methods. For instance, on NQ, CORE achieves a token compression ratio of 3.6% while improving the EM score by 3.2 points compared to the full 10-document input. The results indicate that our method captures **intrinsic compression patterns** rather than overfitting to a specific input length, enabling it to handle significantly larger context windows effectively.

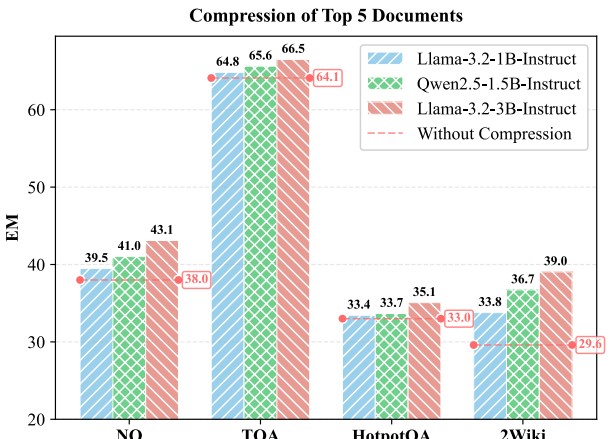

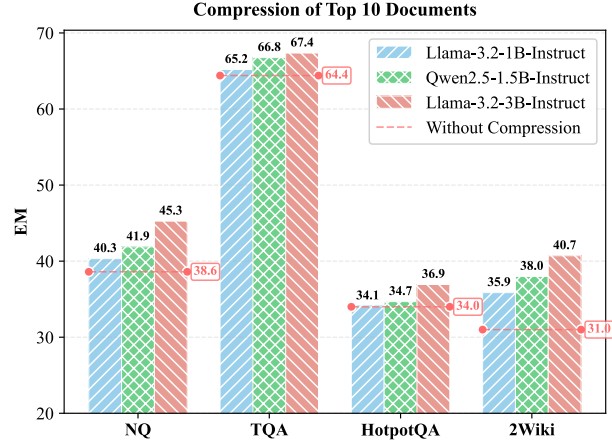

*Figure 5.* Robustness and Scalability: Consistent effectiveness of the proposed method across varying compressor backbones (LLaMA vs. Qwen) and model scales (1B to 3B). More detailed results are provided in Table 7 and Table 8 in the Appendix.

*Table 3.* QA results on four benchmarks with Llama-3.1-8B-Instruct as the LLM ($M$). We report the zero-shot transfer performance on this unseen LLM.

| | NQ | | | TriviaQA | | | HotpotQA | | | 2WikiMultihopQA | | |
|---|---|---|---|---|---|---|---|---|---|---|---|---|
| | EM | F1 | # tok | EM | F1 | # tok | EM | F1 | # tok | EM | F1 | # tok |
| No Retrieval | 24.04 | 34.91 | 0 | 55.64 | 62.57 | 0 | 19.93 | 27.75 | 0 | 27.64 | 31.18 | 0 |
| ***RAG without compression*** | | | | | | | | | | | | |
| Top 1 Document | 33.80 | 44.06 | 142 | 59.17 | 67.50 | 143 | 27.95 | 37.49 | 147 | 28.41 | 33.43 | 153 |
| Top 3 Documents | 36.87 | 47.81 | 427 | 61.13 | 70.06 | 430 | 30.17 | 40.71 | 442 | 28.67 | 34.23 | 460 |
| Top 5 Documents | 37.65 | 48.87 | 712 | 62.26 | 71.04 | 715 | 31.44 | 42.16 | 737 | 29.43 | 35.18 | 766 |
| Top 10 Documents | 38.12 | 49.93 | 1425 | 63.95 | 72.71 | 1428 | 32.19 | 42.62 | 1471 | 30.45 | 36.04 | 1531 |
| ***Compression of top 5 documents*** | | | | | | | | | | | | |
| Qwen2.5-1.5B | 32.60 | 44.21 | 36 | 56.76 | 65.77 | 30 | 26.86 | 36.90 | 33 | 25.45 | 30.88 | 32 |
| DeepSeek-V3 (671B) | 37.56 | 50.11 | 54 | 62.52 | 72.34 | 50 | 33.05 | 44.25 | 48 | 28.64 | 33.87 | 92 |
| RECOMP-Abs (1.5B) | 33.41 | 45.50 | 58 | 58.50 | 67.37 | 53 | 28.85 | 39.76 | 56 | 31.63 | 37.81 | 52 |
| RECOMP-Ext (1.5B) | 33.12 | 45.06 | 60 | 57.98 | 66.84 | 55 | 29.03 | 40.04 | 52 | 31.85 | 38.02 | 55 |
| **CORE (1.5B)** | **40.72** | **50.00** | **46** | **64.08** | **71.13** | **32** | **32.17** | **43.71** | **36** | **35.99** | **41.42** | **49** |
| ***Generalization to top 10 documents (with the compressor trained on top 5 documents)*** | | | | | | | | | | | | |
| Qwen2.5-1.5B | 32.88 | 44.66 | 40 | 57.44 | 66.56 | 33 | 27.31 | 37.31 | 36 | 25.80 | 31.30 | 34 |
| DeepSeek-V3 (671B) | 37.49 | 51.28 | 56 | 63.79 | 73.80 | 53 | 34.24 | 45.35 | 50 | 31.45 | 37.09 | 40 |
| RECOMP-Abs (1.5B) | 34.18 | 46.80 | 59 | 59.69 | 68.89 | 52 | 30.17 | 41.42 | 55 | 33.61 | 39.78 | 44 |
| RECOMP-Ext (1.5B) | 34.06 | 46.55 | 60 | 59.33 | 68.71 | 50 | 30.52 | 41.98 | 55 | 33.52 | 39.42 | 44 |
| **CORE (1.5B)** | **41.77** | **51.27** | **52** | **65.25** | **72.45** | **33** | **33.25** | **45.09** | **37** | **37.59** | **42.87** | **48** |

pronounced when RL is omitted, underscoring the critical role of RL in the absence of ground-truth supervision.

### 4.4. Hyperparameter Study of $\alpha$

Figure 4 illustrates the performance of our method across varying values of $\alpha$, which controls the weighting coefficient of the F1 reward term. Setting $\alpha = 0$ corresponds to opti-mizing solely for the EM reward. The results indicate that setting $\alpha > 0$ outperforms the $\alpha = 0$ setting in the majority of cases across all datasets, demonstrating the effectiveness of the F1 reward in mitigating the sparsity issue inherent to the EM reward. As $\alpha$ increases, performance generally exhibits an initial rise followed by a decline. While the optimal value of $\alpha$ is dataset-dependent, values between 0.2 and 0.5 generally yield robust performance.

*Table 4.* Zero-shot transferability to an unseen dataset (HotpotQA). Both RECOMP and CORE are trained on NQ.

|                              | EM    | F1    | #tok |
|------------------------------|-------|-------|------|
| No Retrieval                 | 21.05 | 29.48 | 0    |
| Top 5 Documents (Full-Context) | 32.99 | 43.69 | 737  |
| RECOMP                       | 29.93 | 41.09 | 45   |
| CORE                         | 33.67 | 45.06 | 36   |
| RECOMP-Transfer              | 26.18 | 37.29 | 52   |
| CORE-Transfer                | 32.15 | 43.26 | 38   |

### 4.5. Robustness and Scalability Across Compressor Models

To evaluate the robustness and scalability of CORE across different model architectures and sizes, we compared the performance of compressors trained using various backbones (LLaMA-3.2-1B-Instruct, Qwen2.5-1.5B-Instruct, and LLaMA-3.2-3B-Instruct) while keeping the downstream LLM (Qwen2.5-14B-Instruct) fixed. As shown in Figure 5, the results indicate that: (1) These trained compressors consistently outperform the full-context baseline (indicated by the red reference line), achieving performance-lossless compression. This result confirms that our training framework is model-agnostic. (2) The performance improves as the size of the compressor model increases (from 1B to 3B), consistent with scaling laws. However, for the sake of efficiency, we avoid using excessively large compressors.

### 4.6. Transferability Across Different LLMs ($M$)

One advantage of textual summaries is their inherent transferability to other LLMs, unlike soft compression approaches (Chevalier et al., 2023; Cheng et al., 2024). We evaluate whether our compressor, which was optimized for a specific LLM (Qwen2.5-14B-Instruct), can effectively transfer to other LLMs. Table 3 presents the results of transferring to LLaMA-3.1-8B-Instruct. As shown, baseline compressors exhibit limited generalization, suffering a significant performance drop compared to the full-context baseline. In stark contrast, our method (CORE) not only matches but surpasses the performance of the full-context baseline when applied to this unseen LLM, all while maintaining a high compression rate. These findings suggest that CORE generates inherently high-quality summaries that capture essential information relevant to the current query, thereby facilitating robust transfer across different LLMs.

### 4.7. Transferability Across Different Datasets

We evaluate whether our compressor, trained on one dataset, can effectively transfer to an unseen dataset. Specifically, we directly apply compressors (RECOMP and CORE) trained on a single-hop QA dataset (NQ) to a multi-hop dataset (HotpotQA). As shown in Table 4, our transferred model achieves nearly lossless performance relative to the full-context baseline, significantly outperforming the RECOMP baseline. While both our method and the baseline underperform compared to versions trained directly on the target HotpotQA dataset, our approach suffers from significantly less degradation. Consequently, our method demonstrates superior robustness and cross-dataset transferability.

### 4.8. Case Study

To conduct an in-depth analysis of our method's advantages, we performed case studies comparing summaries generated by the off-the-shelf Qwen2.5-1.5B-Instruct, RECOMP, and CORE. All summaries were derived from the same document set, and we evaluated the subsequent answers predicted by the LLM when conditioned on these summaries. Due to space limitations, we present two representative cases from the single-hop QA dataset (NQ) and the multi-hop QA dataset (2Wiki) in Table 9 of the Appendix. While the summaries produced by the off-the-shelf Qwen2.5-1.5B are concise, they largely fail to capture the key information required to answer the question. Although RECOMP demonstrates stronger summarization capabilities, it struggles with lengthy documents, leading to misjudgments and misleading hallucinations. In contrast, CORE accurately summarizes answer-critical information, enabling the LLM to generate the correct answer. More details can be found in Section D.4 of the Appendix.

## 5. Conclusion

This paper analyzes the limitations of existing context compression methods for RAG. A primary challenge lies in the absence of optimal reference summaries for supervised learning, which leads to performance degradation in downstream tasks. To address this, we frame compression as a reinforcement learning problem, utilizing task performance as a reward signal to train the compression policy, thereby enabling performance-driven optimization. Extensive experiments demonstrate that our method achieves higher accuracy than standard full-context RAG while significantly reducing context length. Further in-depth analysis provides additional insights into the robustness and transferability.

## Acknowledgements

This work is supported by the Early Career Scheme (No.CityU 21219323) and the General Research Fund (No.CityU 11220324) of the University Grants Committee (UGC), the NSFC Young Scientists Fund (No.9240127), and the Donation for Research Projects (No.9229164 and No.9229216).

## Impact Statement

This paper explores performance-driven context compression for RAG. The goal of this paper is to advance the field of machine learning. We assert that it does not introduce any negative social implications that would necessitate further discussion.

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

## A. Prompt Templates

Figure 3 displays the prompts used for our method. The upper part of the figure illustrates the prompt used to generate a summary of the retrieved documents, conditioned on the given query. Notably, this prompt is concise. For end-task answer generation, the prompt provided to the LLM $M$ is shown in the lower part of Figure 3; it incorporates few-shot in-context examples, the compressed context, and the question.

## B. Efficiency Analysis

### B.1. Training Efficiency

Our training pipeline maintains high efficiency across both stages. The initial distillation phase incurs a computational cost comparable to standard SFT. In the subsequent reinforcement learning phase, despite the typical expense of RL, our process remains highly efficient because it optimizes only the lightweight compressor (e.g., 1.5B parameters) while keeping the large generator LLM (e.g., 14B parameters) fixed. This design incurs significantly lower computational costs compared to methods that require fine-tuning the large generator itself. For instance, RL training typically converges within two epochs, taking approximately 5 hours on eight H20 GPUs. Furthermore, the strong generalization capability of our compressor allows a single trained model to be broadly applied across different LLMs and datasets, minimizing the need for frequent retraining and further reducing overall costs. Crucially, the reinforcement learning is exclusive to the training phase and imposes no overhead during inference.

### B.2. Inference Efficiency

CORE significantly enhances inference efficiency by shifting the heavy processing load from the large generator to the lightweight compressor. Unlike standard RAG approaches where the large generator must encode thousands of document tokens, our lightweight compressor condenses long contexts into only a few dozen tokens. Since the compressor is an order of magnitude smaller than the generator LLM, this substitution drastically reduces total encoding latency.

## C. Implementation Details.

For the distillation phase, we perform full-parameter supervised fine-tuning on the off-the-shelf model (e.g., Qwen2.5-1.5B-Instruct) for two epochs using LLaMA-Factory (Zheng et al., 2024). This warmed-up model serves as the initialization for the subsequent RL phase. For the RL phase, we adopt the Verl framework (Sheng et al., 2025) for RL training. The model is optimized using GRPO for two epochs on each dataset. Experiments are conducted on eight NVIDIA

*Table 5.* Comparison results on the LongBench benchmark, specifically focusing on the three Multi-Doc QA tasks.

| | LongBench | | | | | |
|---|---|---|---|---|---|---|
| | MuSiQue | | HotpotQA | | 2WikiMultihopQA | |
| | F1 score | #token | F1 score | #token | F1 score | #token |
| No Context | 26.49 | 0 | 51.13 | 0 | 43.95 | 0 |
| Full Context | 38.41 | 11214 | 62.22 | 9151 | 57.97 | 4887 |
| LongLLMLingua | 33.18 | 983 | 57.69 | 907 | 53.26 | 489 |
| **CORE** | **38.94** | **129** | **63.58** | **126** | **59.73** | **108** |

H20 GPUs with full-parameter updates, a learning rate of 1e-5, a batch size of 256. We set the number of rollouts per sample to $G = 5$ and the KL divergence coefficient to $\beta = 0.001$. The weight of the F1 reward term, $\alpha$, is tuned over the set $0.2, 0.5, 1.0$ using the validation set. The downstream LLM ($M$) used for reward generation is served via the vLLM inference engine during RL training.

## D. Additional Experimental Results

### D.1. Comparison Results on LongBench

To further demonstrate the efficacy of our approach in extremely long-context scenarios, we incorporated an evaluation using the LongBench (Bai et al., 2024) benchmark. LongBench comprises three English multi-document QA tasks: MuSiQue, HotpotQA, and 2WikiMultihopQA. With average document lengths of 11,214, 9,151, and 4,887 tokens, respectively, these datasets are highly representative of long-context environments. Note that the HotpotQA and 2WikiMultihopQA in LongBench differ from those used in Table 1. The primary distinctions lie in the context length and the methodology for document retrieval and construction. We compared four experimental settings: (1) no document context, where the input consists solely of the question; (2) full document context, utilizing the documents without compression; (3) compression via LongLLMLingua; and (4) compression via our proposed method, CORE. In all settings, we employed Qwen2.5-14B-Instruct as the generator LLM. Both LongLLMLingua and our method were trained using Qwen2.5-1.5B-Instruct on the HotpotQA training set. The results are presented in Table 5. As indicated by the results, utilizing full documents yields significant performance gains compared to the no-context baseline, albeit at the cost of substantially increased context length. Regarding compression baselines, while LongLLMLingua reduces document length, it incurs a performance penalty, with F1 scores dropping by an average of 4–5 points relative to the full-document setting. In contrast, our method, CORE, achieves an exceptional compression rate of approximately 2% while maintaining lossless performance; notably, slight improvements are observed across all datasets. These results on the LongBench benchmark further validate the efficacy of our approach in long-context compression scenarios.

## D.2. Robustness Against Noisy Contexts

To evaluate the robustness of our approach against adversarial retrievals and noisy contexts, we constructed a noisy version of the NQ dataset. For each question, we constructed the input context by combining the top-3 passages retrieved by the DPR retriever with 7 randomly selected passages from the Wikipedia corpus to serve as irrelevant/noisy information. This resulted in a context of 10 passages, which were then shuffled to randomize the order. We then compared the performance of our method against the full-document baseline. Experimental results are presented in the table 6. In the "full documents" setting, the downstream LLM directly uses all these 10 passages to answer the question, whereas in our method, the compressor first summarizes the context, and the LLM then generates an answer based on the compressed content. The model we used was trained on the standard NQ dataset without any such noise augmentation. The results show that our method not only matches but slightly surpasses the performance of using all documents, demonstrating its strong noise resistance and ability to extract key information from cluttered contexts. In addition, we compared our approach with the RECOMP baseline, and our method consistently outperforms it, reaffirming the superior compression capability and robustness of our model. Furthermore, our method achieves a high compression rate, condensing the source content from 1,427 tokens to just 48.

## D.3. Impact of Different Compressor Models on Performance

In our previous experiments, we employed Qwen2.5-1.5B as the initial model to train our compressor. In this section, we utilize two additional models—Llama3.2-1B and Llama3.2-3B—as starting points to train our compressor and the baseline compressor, respectively. The detailed experimental results are presented in Table 7 and Table 8. As shown in the results, our method CORE continues to achieve lossless compression with both models, maintaining a high token compression ratio while exhibiting no performance degradation in terms of EM and F1 score compared to uncompressed RAG. Furthermore, under both new model configurations, our approach consistently outperforms the baseline methods, indicating that its superiority is not dependent on a specific model architecture and thus demonstrates strong robustness. We also observe that our method adheres to a form of scaling law: the compressor trained using the 3B model outperforms the one trained with the 1B model. Specifically, the 1B compressor improves performance by 1–4 EM points over the uncompressed baseline, while the 3B compressor yields gains of 3–9 EM points.

*Table 6.* Evaluation on constructed noisy Natural Questions (NQ).

|  | EM | F1 | #tok |
|---|---|---|---|
| full documents | 35.21 | 45.38 | 1427 |
| RECOMP | 33.29 | 43.90 | 59 |
| **CORE** | **38.19** | **48.85** | **48** |

## D.4. Case Study

To conduct an in-depth analysis of the advantages of our compressor, we performed case studies on one single-hop QA dataset (NQ) and one multi-hop QA dataset (2Wiki), with the results presented in Table 9 and Table 10, respectively. For each case, we compared the summaries generated by off-the-shelf Qwen2.5-1.5B-Instruct, RECOMP, and our method CORE based on the same set of documents, as well as the predicted answers generated by the LLM after prepending these summaries. As shown in the tables, although the summaries produced by off-the-shelf Qwen2.5-1.5B are concise, they largely fail to capture key information relevant to answering the question. In contrast, RECOMP demonstrates better summarization capability but is prone to being overwhelmed by lengthy documents, resulting in misjudgments and even generating misleading information—such as the statement in Table 9: "*The U.S. stopped drafting for the Vietnam War after the Selective Service System was officially abolished in December 1972*"—which leads the downstream LLM to produce the incorrect answer "1972". Our method, CORE, accurately extracts answer-critical information from lengthy documents, exemplified by the summary: "*The U.S. stopped drafting for the Vietnam War in 1973 after announcing the decision by Secretary of Defense Melvin Laird earlier that year*", thereby enabling the LLM to generate the correct answer "1973". This indicates that our compressor, trained with an performance-driven optimization strategy, can produce document summaries that are most helpful for answering the given question while effectively filtering out irrelevant information.

*Table 7.* Open-domain QA results using Qwen2.5-14B-Instruct as the downstream LLM ($M$). The reported token counts represent the length of in-context documents, excluding few-shot examples. RECOMP and our method CORE are both trained using **llama3.2-1B-Instruct**.

| | NQ | | | TriviaQA | | | HotpotQA | | | 2WikiMultihopQA | | |
|---|---|---|---|---|---|---|---|---|---|---|---|---|
| | **EM** | **F1** | **# tok** | **EM** | **F1** | **# tok** | **EM** | **F1** | **# tok** | **EM** | **F1** | **# tok** |
| No Retrieval | 0.2136 | 0.3097 | 0 | 0.5323 | 0.5998 | 0 | 0.2105 | 0.2948 | 0 | 0.2611 | 0.2951 | 0 |
| *RAG without compression* | | | | | | | | | | | | |
| Top1 Document | 0.3446 | 0.4441 | 142 | 0.6082 | 0.6870 | 143 | 0.2920 | 0.3893 | 147 | 0.2679 | 0.3187 | 153 |
| Top3 Documents | 0.3778 | 0.4845 | 427 | 0.6260 | 0.7102 | 430 | 0.3164 | 0.4187 | 442 | 0.2789 | 0.3358 | 460 |
| Top5 Documents | 0.3803 | 0.4916 | 712 | 0.6410 | 0.7248 | 715 | 0.3299 | 0.4369 | 737 | 0.2964 | 0.3521 | 766 |
| Top10 Documents | 0.3867 | 0.5003 | 1425 | 0.6440 | 0.7292 | 1428 | 0.3395 | 0.4488 | 1471 | 0.3104 | 0.3675 | 1531 |
| *Compression of top 5 docs* | | | | | | | | | | | | |
| llama3.2-1B | 0.3147 | 0.4227 | 64 | 0.5552 | 0.6415 | 60 | 0.2648 | 0.3639 | 58 | 0.2498 | 0.3003 | 61 |
| Deepseek-V3 (671B) | 0.3773 | 0.5039 | 54 | 0.6528 | 0.7433 | 51 | 0.3359 | 0.4483 | 48 | 0.2507 | 0.3031 | 45 |
| RECOMP (1B) | 0.3410 | 0.4655 | 57 | 0.6071 | 0.6880 | 48 | 0.2987 | 0.4121 | 49 | 0.3045 | 0.3653 | 33 |
| **CORE (1B)** | **0.3947** | **0.4923** | **47** | **0.6483** | **0.7287** | **43** | **0.3344** | **0.4454** | **45** | **0.3378** | **0.3969** | **34** |
| *Compression of top 10 docs (with the compressor trained on top 5 docs)* | | | | | | | | | | | | |
| llama3.2-1B | 0.3141 | 0.4228 | 62 | 0.5651 | 0.6512 | 58 | 0.2663 | 0.3661 | 56 | 0.2493 | 0.3006 | 61 |
| Deepseek-V3 (671B) | 0.3779 | 0.5107 | 56 | 0.6529 | 0.7445 | 53 | 0.3462 | 0.4569 | 50 | 0.2900 | 0.3464 | 40 |
| RECOMP (1B) | 0.3421 | 0.4661 | 59 | 0.6095 | 0.6917 | 52 | 0.2982 | 0.4105 | 55 | 0.3072 | 0.3681 | 44 |
| **CORE (1B)** | **0.4033** | **0.5033** | **47** | **0.6521** | **0.7296** | **45** | **0.3412** | **0.4500** | **48** | **0.3586** | **0.4162** | **42** |

*Table 8.* Open-domain QA results using Qwen2.5-14B-Instruct as the downstream LLM ($M$). The reported token counts represent the length of in-context documents, excluding few-shot examples. RECOMP and our method CORE are both trained using **llama3.2-3B-Instruct**.

| | NQ | | | TriviaQA | | | HotpotQA | | | 2WikiMultihopQA | | |
|---|---|---|---|---|---|---|---|---|---|---|---|---|
| | **EM** | **F1** | **# tok** | **EM** | **F1** | **# tok** | **EM** | **F1** | **# tok** | **EM** | **F1** | **# tok** |
| No Retrieval | 0.2136 | 0.3097 | 0 | 0.5323 | 0.5998 | 0 | 0.2105 | 0.2948 | 0 | 0.2611 | 0.2951 | 0 |
| *RAG without compression* | | | | | | | | | | | | |
| Top1 Document | 0.3446 | 0.4441 | 142 | 0.6082 | 0.6870 | 143 | 0.2920 | 0.3893 | 147 | 0.2679 | 0.3187 | 153 |
| Top3 Documents | 0.3778 | 0.4845 | 427 | 0.6260 | 0.7102 | 430 | 0.3164 | 0.4187 | 442 | 0.2789 | 0.3358 | 460 |
| Top5 Documents | 0.3803 | 0.4916 | 712 | 0.6410 | 0.7248 | 715 | 0.3299 | 0.4369 | 737 | 0.2964 | 0.3521 | 766 |
| Top10 Documents | 0.3867 | 0.5003 | 1425 | 0.6440 | 0.7292 | 1428 | 0.3395 | 0.4488 | 1471 | 0.3104 | 0.3675 | 1531 |
| *Compression of top 5 docs* | | | | | | | | | | | | |
| llama3.2-3B | 0.3252 | 0.4334 | 60 | 0.5650 | 0.6521 | 59 | 0.2772 | 0.3809 | 58 | 0.2485 | 0.2995 | 60 |
| Deepseek-V3 (671B) | 0.3773 | 0.5039 | 54 | 0.6528 | 0.7433 | 51 | 0.3359 | 0.4483 | 48 | 0.2507 | 0.3031 | 45 |
| RECOMP (3B) | 0.3657 | 0.4912 | 55 | 0.6183 | 0.6920 | 47 | 0.3025 | 0.4238 | 52 | 0.3274 | 0.3806 | 42 |
| **CORE (3B)** | **0.4310** | **0.5234** | **32** | **0.6650** | **0.7306** | **38** | **0.3507** | **0.4736** | **51** | **0.3905** | **0.4474** | **40** |
| *Compression of top 10 docs (with the compressor trained on top 5 docs)* | | | | | | | | | | | | |
| llama3.2-3B | 0.3318 | 0.4359 | 61 | 0.5720 | 0.6588 | 57 | 0.2791 | 0.3854 | 60 | 0.2491 | 0.3011 | 59 |
| Deepseek-V3 (671B) | 0.3779 | 0.5107 | 56 | 0.6529 | 0.7445 | 53 | 0.3462 | 0.4569 | 50 | 0.2900 | 0.3464 | 40 |
| RECOMP (3B) | 0.3682 | 0.4963 | 52 | 0.6205 | 0.6973 | 44 | 0.3077 | 0.4261 | 54 | 0.3312 | 0.3869 | 50 |
| **CORE (3B)** | **0.4526** | **0.5467** | **33** | **0.6736** | **0.7404** | **37** | **0.3693** | **0.4926** | **51** | **0.4071** | **0.4633** | **48** |

*Table 9.* Case study on NQ dataset.

**Question: when did the us stop drafting for the vietnam war? Gold answer: [1973]**

**Top-5 documents**:

last men conscripted, who were born in 1952 and who reported for duty in June 1973. On February 2, 1972, a drawing was held to determine draft priority numbers for men born in 1953, but in early 1973 it was announced by Secretary of Defense Melvin Laird that no further draft orders would be issued. In March 1973, 1974, and 1975, the Selective Service assigned draft priority numbers for all men born in 1954, 1955, and 1956, in case the draft was extended, but it never was. Command Sergeant Major Jeff Mellinger, believed to be the last drafted enlisted ranked.

The Gates Commission issued its report in February 1970, describing how adequate military strength could be maintained without having conscription. The existing draft law was expiring at the end of June 1971, but the Department of Defense and Nixon administration decided the draft needed to continue for at least some time. In February 1971, the administration requested of Congress a two-year extension of the draft, to June 1973. Senatorial opponents of the war wanted to reduce this to a one-year extension, or eliminate the draft altogether, or tie the draft renewal to a timetable for troop withdrawal from Vietnam; Senator.

draft operated from 1940 until 1946 when further inductions were suspended, and its legislative authorization expired without further extension by Congress in 1947. During this time, more than 10 million men had been inducted into military service. However, the Selective Service System remained intact. Scattered opposition was encountered especially in the northern cities where some African-Americans protested the system. The tiny Nation of Islam was at the forefront, with many Black Muslims jailed for refusing the draft, and their leader Elijah Muhammed was sentenced to federal prison for 5 years for inciting draft resistance. Organized draft resistance also developed in.

was the first time any form of student deferment was used. During the Korean War a student carrying at least twelve semester hours was spared until the end of his current semester. The United States breathed easier with the Korean War Armistice on July 27, 1953; however, technology brought new promises and threats. U.S. air and nuclear power fueled the Eisenhower doctrine of "massive retaliation". This strategy demanded more machines and fewer foot soldiers, so the draft slipped to the back burner. However, the head of the SSS, Maj. Gen. Hershey, urged caution fearing the conflict looming in Vietnam. In.

policy of using the draft as force to compel "voluntary" enlistment was unique in U.S. history. Previous drafts had not aimed at encouraging individuals to sign up in order to gain preferential placement or less dangerous postings. However, the incremental buildup of Vietnam without a clear threat to the country bolstered this. Some estimates suggest conscription encompassed almost one-third of all eligible men during the period of 1965–69. This group represented those without exemption or resources to avoid military service. During the active combat phase, the possibility of avoiding combat by selecting their service and military specialty led as many.

| Compressor | Generated summary | Predicted answer |
| --- | --- | --- |
| **Off-the-shell Qwen2.5-1.5B-Instruct** | The US stopped drafting for the Vietnam War in late 1972 after the draft expired at the end of June 1971. | late 1972 |
| **RECOMP** | The U.S. stopped drafting for the Vietnam War after the Selective Service System was officially abolished in December 1972 due to growing public opposition and the expiration of the draft's legal authority in 1971. While the draft was formally ended, some individuals were still consripted into the military | December 1972 |
| **Ours** | The U.S. stopped drafting for the Vietnam War in **1973** after announcing the decision by Secretary of Defense Melvin Laird earlier that year. Although the Selective Service System was later updated to include draft priority numbers, this decision marked the end of the draft's use for national service. | 1973 |

*Table 10.* Case study on 2Wiki dataset.

**Question: Who is Charles Bretagne Marie De La Trémoille's paternal grandfather?**
**Gold answer: [Charles Armand René de La Trémoille]**

**Top-5 documents**:

as at Versailles: he was brigadier of cavalry (January 1709), first gentleman of the King's chamber (June 1709), governor of Thouars (July 1709), and Maréchal de camp (February 1719). His sister Marie Armande Victoire de La Trémoille married Emmanuel Théodose de La Tour d'Auvergne. On 13 April 1706 he married Marie-Madeleine Motier de La Fayette (1691–1717), the daughter of Rene-Armand, marquis de La Fayette and Marie-Madeleine de Marillac, and granddaughter of the author Marie-Madeleine Pioche de la Vergne, comtesse de la Fayette. They had one child, Charles Armand René de La Trémoille, born in 1708. Charles Louis Bretagne de La

Charles Bretagne Marie de La Trémoille Charles Bretagne Marie de La Trémoille (24 March 1764 – 10 November 1839), 8th duc de Thouars, was a French soldier and the son of Jean Bretagne Charles de La Trémoille and his wife, Marie-Maximilienne, princess of Salm-Kyrburg. La Trémoille married Louise-Emmanuelle de Châtillon in 1781. She was a grand daughter of Louis César de La Baume Le Blanc, the famous writer. The couple had one daughter: At the outbreak of the French Revolution, he sided with King Louis XVI, as did the rest of his family. In 1789, La Trémoille and his parents

Jean Bretagne Charles de La Trémoille Jean-Bretagne-Charles de La Trémoille (5 February 1737 – 19 May 1792), 7th duc de Thouars, was a French soldier and the son of Charles Armand René de La Trémoille and his wife, Marie Hortense de La Tour d'Auvergne. La Trémoille began his military career in the Seven Years' War as the head of the regiment of Aquitaine-Cavalry. He was seriously wounded and afterwards promoted to the rank of sergeant, and later of brigadier, of the armies of the king. During the French Revolution, La Trémoille remained faithful to King Louis XVI. He emigrated in

Charles Louis Bretagne de La Trémoille Charles Louis Bretagne de La Trémoille (15 March 1683 – 9 October 1719), 6th duke of Thouars, was the son of Charles Belgique Hollande de La Trémoille and Madeleine de Créquy, daughter and heiress of Charles III de Créquy. He became duke of Thouars upon his father's death in 1709. He was also count of Laval and Montfort and inherited the ancient French claim to the Kingdom of Naples. The Neapolitan claim gave the family the rank of "princes étrangers" at the French court. The Duke made a career in the military as well

de Saint-Didier. They had two daughters: Following her death, La Trémoille married for a third time in 1830 to Valentine Eugénie Joséphine Walsh de Serrant. They had two children: La Trémoille died shortly after his son's birth, in 1839. Charles Bretagne Marie de La Trémoille Charles Bretagne Marie de La Trémoille (24 March 1764 – 10 November 1839), 8th duc de Thouars, was a French soldier and the son of Jean Bretagne Charles de La Trémoille and his wife, Marie-Maximilienne, princess of Salm-Kyrburg. La Trémoille married Louise-Emmanuelle de Châtillon in 1781. She was a grand daughter of Louis César de

| Compressor | Generated summary | Predicted answer |
|---|---|---|
| **Off-the-shell Qwen2.5-1.5B-Instruct** | Charles Bretagne Marie de La Trémoille's paternal grandfather was Jean Bretagne Charles de La Trémoille. | Jean Bretagne Charles de La Trémoille |
| **RECOMP** | Charles Bretagne Marie de La Trémoille's paternal grandfather was Charles Louis Bretagne de La Trémoille, who was the son of Charles Belgique Hollande de La Trémoille and Madeleine de Créquy. This makes him the direct paternal grandfather of Charles Bretagne Marie de La Trémoille. | Charles Louis Bretagne de La Trémoille |
| **Ours** | Charles Bretagne Marie de La Trémoille's paternal grandfather is Charles Armand René de La Trémoille, her father's father, the 7th duc de Thouars. | Charles Armand René de La Trémoille |

