# OpenReview forum: "Less Is More: Elevating RAG via Performance-Driven Context Compression"
_ICML.cc/2026/Conference — ICML 2026 regular_

### Official Review · Reviewer_u8JZ · 2026-03-05

**Soundness:** 4
**Presentation:** 3
**Significance:** 3
**Originality:** 3
**Overall Recommendation:** 5
**Confidence:** 4

**Summary:**

This paper proposes CORE-RAG, a performance-driven RAG context compression framework with distillation warm-up and RL optimization. It achieves 3-6% compression, outperforming full-context RAG and SOTA baselines on QA tasks, with strong transferability.

**Compliance With Llm Reviewing Policy:**

Affirmed.

**Key Questions For Authors:**

1. This paper provides an excellent RAG content filtering module—but we have observed that large-scale models already possess this capability. I wonder if this filtering function remains effective for even larger-scale models?
2. Does the supervised content filtering function suffer from poor generalizability? Could it increase hallucinations?

**Limitations:**

yes

**Strengths And Weaknesses:**

### Strengths
1. **Soundness**: Formally defines RAG context compression with rigorous RL optimization (GRPO) and composite EM/F1 rewards. Validated across 4 datasets, 3 compressor backbones, and 2 LLMs, with ablation studies confirming distillation/RL necessity. Robust to noise and long contexts (LongBench) with consistent performance gains.
2. **Presentation**: Clearly structures methodology (distillation + RL) and experiments, with intuitive visualizations (performance-token tradeoffs) and detailed tables. Appendices provide prompt templates, efficiency analysis, and case studies, enhancing reproducibility.
3. **Significance**: Solves RAG’s key pain points (computational cost, "lost-in-the-middle") via 3-6% compression while improving accuracy. Lightweight, plug-and-play design enables easy integration; strong transferability across LLMs/datasets boosts practical deployment value.
4. **Originality**: First performance-driven RAG compression framework, eliminating proxy heuristics. Innovates two-stage training (distillation warm-up + RL) and generative compression, outperforming heuristic/binary token-dropping methods.

### Weaknesses
1. **Soundness**: Limited to QA tasks; untested on summarization/dialogue. No analysis of moderate compression ratios (20-50%) or performance on small LLMs (<7B). Relies on strong retrievers, with unproven robustness to low-quality retrieval.
2. **Presentation**: Lacks human evaluation of summary readability/factual consistency. GRPO algorithm choice is not justified vs. PPO. No discussion of compressor architectural design tradeoffs (e.g., prompt template impact).
3. **Significance**: Untested on large-scale knowledge bases (million+ chunks) or high-concurrency scenarios. No cost-benefit comparison with lightweight heuristics (e.g., keyphrase extraction) for resource-constrained environments.
4. **Originality**: Builds on standard RL (GRPO) and distillation techniques without mathematical extensions. Compression logic lacks interpretability—no explanation of how critical information is prioritized.

---

> ### Author Rebuttal · Authors · 2026-03-31
>
> **Hi, Reviewer u8JZ:**
>
> Thank you for acknowledging CORE’s **rigorous formulation, methodological originality, extensive evaluation, lightweight plug-and-play design, strong transferability, clear presentation, and high reproducibility**.
>
> Below are our detailed responses.
>
> **`Re. to Q1`**
>
> - **Large-scale LMs still lack robust content filtering capabilities for long contexts.** Recent studies [1, 2] show **even strong LLMs, such as GPT-3.5/4o and Claude, suffer severely from the "lost in the middle" phenomenon, where critical information is drowned out by surrounding noise**. Therefore, CORE remains both necessary and effective for larger-scale models by proactively filtering noise before it reaches the LLM.
>
> [1] Liu, et al. Lost in the middle: How language models use long contexts. TACL (2024)
>
> [2] Zhang, et al. Lost-in-the-Middle in Long-Text Generation: Synthetic Dataset, Evaluation Framework, and Mitigation. (2025)
>
> **`Re. to Q2`**
>
> - **Generalizability**: CORE demonstrates **strong generalization across 2 downstream LLMs (Sec 4.6), different datasets (Sec 4.7), and 3 compressor backbones (Sec 4.5)**. Furthermore, RL training inherently helps improve generalization compared to pure SFT.
>
> - **Hallucination: CORE mitigates, rather than increases, hallucinations**. Our RL reward strictly penalizes ungrounded answers, forcing the compressor to retain exact factual evidence. Our case studies (Appendix Tables 9&10) confirm the compressed contexts are highly faithful to source texts without introducing hallucinations.
>
> **`Re. to W1`**
>
> - **Task**: We focus on QA because it is the most widely adopted and representative task for RAG, and it aligns with our baselines’ experimental settings. However, CORE is fundamentally task-agnostic. For discussion on adapting it to summarization or dialogue tasks, please refer to our **response to Reviewer bpDj’s W2**.
>
> - **Compression ratio: Our method achieves extreme compression (generating a compressed context that is only *3%* the length of the original**), directly addressing RAG's critical computational bottleneck. Success at this extreme level strongly implies effectiveness at easier, moderate ratios (20-50%).
>
> - **Retriever: Our method does not rely on strong retrievers** because: (1) We evaluated across three different retrievers (Sec. 4.1.3); and (2) we explicitly evaluate robustness under low-quality retrieval by injecting noisy contexts (Appendix D.2).
>
>
> **`Re. to W2`**
>
> - **Human evaluation**: While human evaluation is a valuable suggestion, the compressed context in our framework is intended for the downstream LLM rather than human users. Therefore, downstream QA metrics are the most direct measure of its utility. Case studies (Appendix Tables 9&10) verify readability and factual consistency of our generated summaries.
>
> - We chose GRPO over PPO for computational efficiency and stability, avoiding the separate value model required by PPO.
>
> - **Prompt**: We deliberately utilized simple and straightforward prompt templates following baselines (RECOMP) to minimize the influence of prompt engineering. This ensures the improvements stem from our performance-driven training framework, not prompt tuning.
>
> **`Re. to W3`**
>
> - **Large-scale knowledge bases**: The size of the knowledge base (e.g., million+ chunks) does not affect the compressor. In a standard RAG pipeline, the compressor operates post-retrieval. The retriever first narrows down the massive knowledge base to a small set of top-$k$ documents. The compressor only processes this retrieved subset, making its performance and latency independent of total knowledge base size.
>
> - **High-concurrency scenarios**: Built on standard LLM architectures (e.g., Qwen, Llama), CORE is fully compatible with highly optimized serving engines like vLLM, inherently resolving high-concurrency challenges.
>
> - **Comparison with Lightweight Heuristics**: In Table 1 of our paper, we evaluated this trade-off by including BM25, a classic heuristic extraction baseline. The results show BM25 performs significantly worse. While cheap, heuristics lack reasoning, matching only surface tokens. They fail to integrate scattered evidence, outputting fragmented snippets that hinder downstream LLM reasoning.
>
> **`Re. to W4`**
>
> - While building on standard RL and distillation, our core contribution is a novel performance-driven training framework. Rather than relying on human-annotated summaries or heuristic proxies, we directly align the compressor's output with the downstream LLM's actual performance, offering a system-level innovation for RAG.
>
> - **Interpretability**: Critical information prioritization is fundamentally governed by our reward mechanism. Optimizing directly for downstream metrics explicitly teaches the compressor to retain essential evidence. Furthermore, case studies (Tables 9 & 10) provide concrete interpretability by illustrating exactly how the compressor extracts critical information and discards noise.

---

> > ### Author Rebuttal · Reviewer_u8JZ · 2026-04-01
> >
> > thanks for your response!

---

> > > ### Author Response · Authors · 2026-04-02
> > >
> > > Dear Reviewer u8JZ,
> > >
> > > We are very glad to hear that our response has fully addressed all of your concerns.
> > >
> > > Thank you very much for your time and for reviewing our rebuttal.

---

### Official Review · Reviewer_XqqM · 2026-03-13

**Soundness:** 3
**Presentation:** 3
**Significance:** 2
**Originality:** 2
**Overall Recommendation:** 5
**Confidence:** 4

**Summary:**

The paper proposes a new method for context compression that can be used for retrieval-augmented generation. The proposed method leverages RL training to optimize a policy model that can compress retrieved documents into shorter contexts for specific downstream tasks. The experiments show strong empirical results against existing context compression baselines across a wide range of QA datasets, and further demonstrate the generalizability of the proposed method across different datasets and numbers of documents.

**Compliance With Llm Reviewing Policy:**

Affirmed.

**Final Justification:**

The clarification and the new baseline results with DAC in the rebuttal have addressed my concerns. I have updated my score accordingly.

**Key Questions For Authors:**

1. Section 3.2.1 — Data Construction: Are the training examples balanced between the two cases described in this section? What does the distribution of examples look like?

2. Section 3.2.1 — Training Objective: Why was DPO not considered as an alternative objective? For example, summaries where
𝑝_{summary} > 𝑝_{original} could be treated as positive examples, while summaries where
𝑝_{summary} < 𝑝_{original}
 could be treated as negative examples for the same instance.

3. GRPO Training and Figure 2: For each rollout that generates a summary for the same example, is a single answer sampled from the frozen model conditioned on the question and the summary, or are multiple answers sampled? Section 3.2.3 seems to indicate the former, while Figure 2 appears to suggest the latter.

4. Figure 5 in Appendix A: Why are summaries restricted to two sentences? Is there a particular reason not to allow summaries with dynamically determined lengths?

**Limitations:**

Yes

**Strengths And Weaknesses:**

Strength

1. The method demonstrates strong performance compared to baseline compression approaches.

2. The results show good transferability across different datasets and show that the trained compressor generalizes well to out-of-domain tasks.

3. The method also generalizes across different numbers of documents in the context.

4. The approach can be extended to much longer context settings, as demonstrated by the evaluation results on LongBench.

Weakness
1. The idea of using RL to train a policy for context compression, optimized based on downstream task performance, is not entirely new in the literature. In particular, TACO (Shandilya et al., 2025) is discussed in Section 2.2 as being substantially different from the proposed method. However, the main differences I observe between TACO and the proposed method appear to be the RL algorithm (TACO uses REINFORCE) and the output format of the policy model (TACO generates binary labels for token selection). It is therefore unclear how much technical novelty this paper contributes relative to TACO and other related works with similar ideas (e.g., EffComp [1]).

2. The proposed method should also be compared with more recent baselines in the literature, for example, methods published in 2025 such as DAC [2] and CoLoR [3].

---

> ### Author Rebuttal · Authors · 2026-03-31
>
> **Hi, Reviewer XqqM:**
>
> Thank you for acknowledging CORE’s **strong performance, good transferability, generalization across different numbers of retrieved docs, and its ability to extend to much longer-context settings**.
>
> Below are our detailed responses to your concerns.
>
> **`Re. to W1: Distinction from TACO`**
>
> - A fundamental distinction is that **CORE is task-performance-driven:** it directly optimizes compressed contexts for downstream task success. In contrast, **TACO is similarity-driven and output-matching:** Its *“task-aware”* reward is defined by the similarity between downstream LLM outputs under the original and compressed prompts, rather than by gold-answer-based task reward.
>
> - If TACO were adapted to the RAG setting, it would reward the compressor **when the downstream LLM’s output based on compressed retrieved documents is similar to its output based on full retrieved documents.** Thus, its optimization target is the full-context system output, not the gold answer. CORE, by contrast, directly **optimizes compressed context against gold supervision to improve end-task performance**. This distinction is especially important in RAG, where **the full retrieved context is often noisy and redundant**. In such cases, the downstream LLM’s output from the full context can itself be influenced by irrelevant or distracting information. As a result, **matching the full-context system’s output is not equivalent to maximizing answer correctness**. This also helps explain the empirical difference: **TACO's compression degrades downstream task performance, whereas CORE's compression not only preserves task performance but also improves it.**
>
> - Beyond this fundamental distinction, **CORE and TACO also differ in several other important dimensions**, as summarized below.
>
> |Dimension|TACO|CORE (Ours)|
> |-|-|-|
> |Problem setting|General prompt compression|Retrieved-context compression for RAG|
> |Input to compressor|Prompt|Question+retrieved docs|
> |Reward|Output similarity between the original-prompt and compressed-prompt outputs|Gold-answer-based task metric|
> |Optimization target|Similarity-driven, behavior-preserving|Performance-driven|
> |Optimization algorithm|REINFORCE|GRPO|
> |Compression mechanism|Binary token selection (Select a subset of original tokens)|Generative summarization (Can rewrite, abstract, and synthesize evidence across docs)|
> |Compressor architecture|Bidirectional encoder + classifier|Lightweight LLM|
> |Compression level (token retention; lower is more aggressive)|16%-50%|3%–6%|
> |Performance|Underperforms the uncompressed-input setting|Surpasses uncompressed-input setting|
>
> **`Re. to W2: New baseline`**
>
> Following the suggestion, **we added DAC [1] as a new baseline**.
>
> |Dataset|Method|EM|F1|#tok|
> |-|-|-|-|-|
> |NQ|DAC|36.51|46.39|60|
> |NQ|CORE|41.02|50.40|46|
> |Hotpot|DAC|28.95|40.24|48|
> |Hotpot|CORE|33.67|45.06|36|
>
> These results show that CORE consistently outperforms DAC on both datasets. We will include these results in Table 1 and expand Related Work to discuss DAC and CoLoR.
>
> **`Re. to Q1`**
>
> We provide the distribution statistics of the distillation training data in the below table:
> |Dataset|Case1|Case2|
> |-|-|-|
> |NQ|34870|4079|
> |TQA|22149|5318|
> |Hotpot|30920|4786|
> |2Wiki|17432|12797|
>
> The two cases are not balanced; Case 1 is more frequent than Case 2. This is expected because Case 2 is defined by a stricter condition: the original setting must already answer correctly, and adding the summary must hurt performance.
>
> **`Re. to Q2`**
>
> - We agree that DPO is meaningful when reliable preference pairs are available. However, in our framework, Sec. 3.2.1 is only a warm-start stage, for which **SFT is more stable and efficient**, while the actual **task-aligned optimization is performed later by online GRPO** using task rewards.
>
> - In addition, **our distillation data do not naturally form standard DPO pairs**: DPO requires a chosen response and a rejected response for the same input, whereas **for each input $(q, D)$ we only have one teacher summary**, and compare it against the absence of a summary rather than a rejected summary.
>
> **`Re. to Q3`**
>
> We apologize for the ambiguity in Fig. 2. For each training sample, we sample a group of summaries; each sampled summary is sent once to the LLM and produces **one corresponding answer**. The multiple answers in the figure denote the one-to-one outputs from the $G$ summary rollouts.
>
> **`Re. to Q4`**
>
> The “two-sentence” setting follows RECOMP [2], our most directly related baseline. We adopted the same prompt format to ensure a fair comparison. Empirically, we found that this prompt already worked well and also made it easy to keep the output length within a certain range, so we did not further explore alternative prompt designs. We leave such extensions to future work.
>
> [1] DAC: A Dynamic Attention-aware Approach for Task-Agnostic Prompt Compression
>
> [2] RECOMP: Improving Retrieval-Augmented LMs with Compression and Selective Augmentation

---

> > ### Author Rebuttal · Reviewer_XqqM · 2026-04-03
> >
> > Thank you for the clarification and the new baseline results with DAC. I have updated my score accordingly.

---

> > > ### Author Response · Authors · 2026-04-04
> > >
> > > Dear Reviewer XqqM,
> > >
> > > We sincerely appreciate your recognition that our clarifications have adequately addressed your concerns.
> > >
> > > Thank you very much for your positive feedback and for raising your score.

---

### Official Review · Reviewer_sCqZ · 2026-03-13

**Soundness:** 3
**Presentation:** 3
**Significance:** 2
**Originality:** 2
**Overall Recommendation:** 3
**Confidence:** 5

**Summary:**

The authors explore context summarization for RAG. Specifically, they summarize all retrieved documents together into a short blurb around 50 tokens, and evaluate up to 5 and even 10 retrieved chunks. Their model is trained via RL to provide summarizes that improve answer quality, and yields empirically strong results across multiple wikipedia-based datasets, matching or exceeding larger untrained models.

**Compliance With Llm Reviewing Policy:**

Affirmed.

**Final Justification:**

Thanks for the response. I am improving my score for soundness and presentation, but still lean reject.

I want to emphasize that there are a huge number of non-wikipedia retrieval datasets available including very popular ones such as BEIR, BRIGHT, and the ones in RTEB. Additionally, wikipedia is heavily used for model training, and models can probably answer many of the NQ and HPQA questions w/o retrieval, severely limiting the value of the empirical results.

**Key Questions For Authors:**

n/a

**Strengths And Weaknesses:**

S1. The paper explores extreme context compression, which will be a big cost saver when answer generation is expensive. Their trained model also gives a quality boost, so it is on the pareto frontier.

S2. The results include many models to compare with. It does seem the trained 1.5b yields similar benefits as much larger models.

S3. The analysis and ablations provide a thorough investigation of the robustness properties of CORE.

W1. The tables are sometimes difficult to read. The authors' result is bolded, but it is more typical for the max or min result per column to be bolded.

W2. The data generalization claims are weak -- NQ and HPQA are both wikipedia based.

W3. The approach is not very scalable because it compresses all documents at once. It would be interesting to see the impact of a more scalable approach even if it does not reduce token count as much, e.g. summarizing each doc separately as done in ALCE.

W4. I am not sure this approach would hold up well on more complex datasets such as SEMQA or any dataset where the chunks included tables. Plus, even though we are using a small model, it's likely been exposed to wikipedia data which gives a large advantage.

---

> ### Author Rebuttal · Authors · 2026-03-31
>
> **Hi, Reviewer sCqZ:**
>
> Thank you for acknowledging CORE’s **cost-saving extreme compression, strong empirical performance**, and its position on the **Pareto frontier** of cost and quality, as well as the **thorough investigation of its robustness**.
>
> Below, we provide responses to your concerns.
>
> **`Re. to W1: Boldface formatting in tables`**
>
> Thank you for this helpful suggestion. We agree that bolding our method row can sometimes make the tables harder to read. In the revision, we will adopt the conventional formatting of bolding the best result in each column to improve readability.
>
> That said, we would like to clarify that this formatting choice does not affect the substantive conclusion of the tables.
> - **For the EM and F1 columns in Table 1, the bolded CORE entries are in fact always the best results** in their respective columns.
> - For the token-count columns, CORE achieves the best token efficiency among trainable compression baselines. Beyond the trainable baselines, BM25 and off-the-shelf Qwen2.5-1.5B can sometimes yield slightly shorter outputs, but they perform substantially worse in EM/F1.
>
> **`Re. to W2: Dataset generalization`**
>
> We agree that NQ and HotpotQA are built on Wikipedia. Our intention in Sec. 4.7 is to evaluate **cross-dataset transferability under distribution shift**. We would like to emphasize that **this transfer setting is still non-trivial** for several reasons:
> - **Different reasoning requirements**: NQ is single-hop, whereas HotpotQA is multi-hop and requires aggregating evidence across multiple passages.
> - **Different retrieved-context distributions due to different retrievers**: In our experiments, NQ contexts are retrieved with the DPR retriever, while HotpotQA docs are retrieved with Contriever. Thus, the transferred compressor faces retrieval-induced distribution shift.
> - **Meaningful relative robustness**: Under this same transfer setting, our method degrades much less than the baseline method. Therefore, even within Wikipedia, the results still support stronger robustness to dataset shift than baseline methods.
> We agree that evaluating transfer on non-Wikipedia QA would further broaden the empirical scope. Since most widely used open-domain QA benchmarks are Wikipedia-based, non-Wikipedia alternatives are less readily available. Due to time constraints, we leave this extension to future work.
>
>
> **`Re. to W3: Scalability; Joint vs. Separate Document Compression`**
>
> - **Our design uses joint compression intentionally**, rather than compressing each doc separately, because in multi-doc QA the answer often depends on **cross-doc reasoning and evidence aggregation**. Summarizing docs independently can miss such interactions and preserve redundant information that could otherwise be removed when the retrieved set is compressed jointly.
> - **In our preliminary experiments, we directly compared the two compression strategies. The comparison results on HotpotQA (shown in the table below) indicate that joint comp performs better than document-wise comp.**
> ||EM|#tok|
> |-|-|-|
> |Separate comp|31.49|64|
> |Joint comp (CORE)|33.67|36|
>
> - We also clarify that CORE doesn’t need to compress an unbounded set of passages: it operates only on the top-$K$ documents returned by the retriever, where $K$ is typically not very large in standard RAG. Moreover, the compressor is lightweight, so jointly processing a long retrieved context at the compression stage is still substantially more efficient than having the downstream LLM consume the same full uncompressed context directly.
> - Table 1 also shows that CORE scales to larger retrieved sets: trained on top-5 documents, it generalizes to top-10 without retraining and still outperforms full-context RAG.
>
> **`Re. to W4: Performance on table-containing datasets`**
>
> To directly address this concern, **we have added experiments on HybridQA**, a large-scale multi-hop QA benchmark **consisting of both structured tabular data and textual passages**.
>
> **Our results on the HybridQA dev set show that CORE remains effective in this more challenging setting**. Despite substantially compressing the context, CORE matches or even outperforms the full-context setting.
>
> || EM | F1 | # tok |
> |-|-|-|-|
> | full-context | 54.50 | 63.23 | 829 |
> | **CORE** | **54.79** | **63.36** | **58** |
>
> **Regarding the reviewer’s concern about possible Wikipedia exposure**: We agree that Wikipedia exposure cannot be fully ruled out for pretrained language models. However, even if such exposure exists, it is not a potential advantage specific to CORE, since all methods are compared under identical LLM and retrieval corpus settings. In addition, we follow exactly the same experimental setup as prior baselines, using the same datasets and retrieval corpus. Therefore, any potential exposure applies equally to all methods, and the gains of CORE should mainly reflect the effectiveness of the compression strategy rather than differences in external knowledge.

---

> > ### Author Rebuttal · Reviewer_sCqZ · 2026-04-04
> >
> > Thanks for the response. I am improving my score but still lean reject.
> >
> > I want to emphasize that there are a huge number of non-wikipedia retrieval datasets available including very popular ones such as BEIR, BRIGHT, and the ones in RTEB. Additionally, wikipedia is heavily used for model training, and models can probably answer many of the NQ and HPQA questions w/o retrieval, severely limiting the value of the empirical results.

---

> > > ### Author Response · Authors · 2026-04-05
> > >
> > > Thank you for your continued feedback. We would like to further clarify your concerns.
> > >
> > > ### **`1. The retrieval corpus (Wikipedia) and evaluation datasets used in our experiments are standard settings adopted by our baselines and related top-tier RAG literature.`**
> > >
> > > We respectfully emphasize that **our choice of Wikipedia as the retrieval corpus and our selection of datasets strictly follow the standard settings established by our baselines and recent top-tier RAG papers**.
> > >
> > > To illustrate this, we have summarized the evaluation settings of prior works in the table below (**the first three rows are the direct baselines compared in our paper**):
> > >
> > > |Method (Paper)|Retrieval Corpus Used &nbsp;&nbsp;|Datasets Evaluated|
> > > |-:|:-:|:-|
> > > |**RECOMP** [1]|Wikipedia|NQ, TriviaQA, HotpotQA|
> > > |**NoiseFilter-IB** [2]|Wikipedia|NQ, TriviaQA, HotpotQA|
> > > |**QGC** [3]|Wikipedia|NQ, TriviaQA, HotpotQA|
> > > |HippoRAG [4]|Wikipedia|HotpotQA, 2Wiki, MuSiQue|
> > > |xRAG [5]|Wikipedia|NQ, TriviaQA, WebQA, HotpotQA, TruthfulQA|
> > > |CLM [6]|Wikipedia|NQ, TriviaQA, PopQA, 2Wiki|
> > > |ReSearch [7]|Wikipedia|HotpotQA, 2Wiki, MuSiQue, Bamboogle|
> > > |**CORE (Ours)**|**Wikipedia**|**NQ, TriviaQA, HotpotQA, 2Wiki, MuSiQue**(in Appendix)|
> > >
> > > To ensure a fair comparison with existing compression methods, it is imperative that **we use the exact same retrieval corpus and benchmarks**. Because our setting perfectly aligns with the mainstream standards of the field, our empirical results are highly **reliable and convincing**.
> > >
> > > [1] RECOMP: Improving Retrieval-Augmented LMs with Compression and Selective Augmentation, ICLR 2024
> > >
> > > [2] An Information Bottleneck Perspective for Effective Noise Filtering on Retrieval-Augmented Generation, ACL-long 2024
> > >
> > > [3] LongLLMLingua: Accelerating and Enhancing LLMs in Long Context Scenarios via Prompt Compression,  ACL-long 2024
> > >
> > > [4] HippoRAG: Neurobiologically Inspired Long-Term Memory for Large Language Models, NeurIPS 2024
> > >
> > > [5] xRAG: Extreme Context Compression for Retrieval-augmented Generation with One Token, NeurIPS 2024
> > >
> > > [6] Retrieval is Not Enough: Enhancing RAG Reasoning through Test-Time Critique and Optimization, NeurIPS 2025
> > >
> > > [7] ReSearch: Learning to Reason with Search for LLMs via Reinforcement Learning,  NeurIPS 2025
> > >
> > > ### **`2. The LLM’s potential pre-training exposure to Wikipedia does not limit the value of our empirical results.`**
> > >
> > > Regarding the reviewer’s concern that "*models can probably answer many of the NQ and HPQA questions w/o retrieval, severely limiting the value of the empirical results*", **we respectfully argue that there is no causal relationship here**.
> > > - First, we have already provided a direct comparison between RAG and the "No-Retrieval" setting in Table 1 of our paper. The results clearly show **relying solely on the LLM's internal memory is highly insufficient**. For example, **on the NQ dataset, using "full-context 10doc RAG" improves the Exact Match (EM) score by **81%** compared to the "No-Retrieval" baseline**. This massive improvement proves that retrieving from Wikipedia corpus is highly effective and necessary, regardless of any potential pre-training exposure.
> > > - Second, the core objective of our paper is to evaluate the effectiveness of context compression, not to prove the necessity of retrieval itself (although our baseline results clearly confirm this premise). Our experimental logic is clear and well-supported: **1) Traditional full-context RAG is significantly better than No-Retrieval on all datasets.** $\rightarrow$ **2) Our method CORE exceeds full-context RAG using significantly fewer tokens, while also significantly outperforming other compression baselines.** The fact that CORE achieves better task performance with significantly compressed context fully validates the core claim of our paper.
> > >
> > > ### **`3. Additional experiments on non-Wikipedia datasets (BEIR / BioASQ).`**
> > >
> > > **Despite that current experiments on standard benchmarks are comprehensive and rigorously follow prior work, we are more than happy to provide additional results on a non-Wikipedia dataset to further alleviate the reviewer’s concerns.**
> > >
> > > We conducted new experiments on the BioASQ dataset from the BEIR benchmark. BioASQ is a specialized biomedical dataset that uses PubMed articles as its retrieval corpus, which is strictly out-of-domain from Wikipedia and requires highly specialized domain knowledge. We compared CORE against No-Retrieval, full-context RAG, and the strongest compression baseline, RECOMP.
> > >
> > > - **Comparison Results on BEIR/BioASQ**
> > > |Method|EM|F1|#tok|
> > > |-|-|-|-|
> > > |No Retrieval|12.62|24.33|0|
> > > |Full-context RAG|57.85|76.78|975|
> > > |RECOMP|51.70|71.24|59|
> > > |**CORE (Ours)**|**58.43**|**77.52**|**56**|
> > >
> > > As shown in the table, **our core conclusions remain perfectly consistent** in this non-Wikipedia, highly specialized domain: Full-context RAG significantly outperforms No-Retrieval, and our proposed CORE-RAG outperforms both Full-context RAG and the RECOMP baseline while drastically reducing the token count.

---

### Official Review · Reviewer_bpDj · 2026-03-14

**Soundness:** 2
**Presentation:** 3
**Significance:** 2
**Originality:** 2
**Overall Recommendation:** 3
**Confidence:** 3

**Summary:**

This paper introduces CORE-RAG, a framework for compressing retrieved documents in retrieval-augmented generation. The main motivation is that many existing context compression approaches depend on proxy signals such as lexical overlap, mutual information, or imitation of stronger models, rather than optimizing for the actual downstream objective. The authors argue that this mismatch can lead to suboptimal compression and, in some cases, even hurt end-task performance.
To address this, the paper formulates context compression as a reinforcement learning problem driven by downstream QA performance. The method has two training stages. First, the compressor is warm-started through knowledge distillation using summaries generated by DeepSeek-V3. Then, it is further optimized with GRPO, using the final QA performance of a frozen black-box LLM as the reward. The reward is computed from exact match and token-level F1. The compressor itself is relatively small, based on Qwen2.5-1.5B, while the downstream model is a fixed Qwen2.5-14B. Experiments on NQ, TriviaQA, HotpotQA, and 2WikiMultihopQA show that the method can compress the top-5 retrieved documents into short summaries while outperforming not only other compression baselines but also full-context RAG.

**Compliance With Llm Reviewing Policy:**

Affirmed.

**Key Questions For Authors:**

I also think the paper should say more about training cost and reproducibility. The warm-start stage depends on DeepSeek-V3 as the teacher, and the RL stage requires multiple rollouts together with a 14B LLM in the loop, so the full cost is likely nontrivial. Appendix B reports about 5 hours of training on 8 H20 GPUs, but that figure does not seem to include the cost of generating the teacher summaries. An ablation with smaller teacher models would make the practicality claim easier to evaluate. Since the method relies on RL, it would also be useful to report seed variance, reward trajectories, and some indication of convergence stability.

**Limitations:**

yes

**Strengths And Weaknesses:**

Strengths:

One thing the paper does well is align the method closely with the problem it is trying to solve. The central point is simple and reasonable: what matters for context compression is not whether the summary looks good on its own, but whether it actually helps the downstream model answer the question. Using final task performance as the reward follows naturally from that motivation, and it addresses a real weakness of proxy-based compression methods.

The overall setup is also fairly practical. The downstream LLM is kept fixed, and only a much smaller compressor is trained. That makes the approach easier to imagine as a drop-in component in a real RAG pipeline, without having to retrain or modify the main model.

The experimental section is fairly comprehensive. The paper compares against a broad set of baselines across four QA benchmarks, including full-context RAG, BM25, several summarization baselines, and prior compression methods such as RECOMP, NoiseFilter-IB, LongLLMLingua, and QGC. It also includes transfer from top-5 to top-10 retrieved documents, experiments with different compressor backbones and model sizes, transfer to unseen LLMs, transfer to unseen datasets, and ablations on the distillation and RL stages. That gives the paper more credibility than a result that only works in one narrow setting. The result that stands out most is that CORE can outperform full-context RAG while using a much shorter context, which makes the contribution more interesting.

Another positive aspect is that the method uses text-based hard compression. Since the output is still readable text rather than latent representations or soft masks, it is easier to believe that the compressed output can transfer across different downstream models. The transfer results on unseen LLMs and datasets give some support to that claim.

I also thought the data curation strategy in the distillation stage was sensible. Keeping only teacher summaries that actually improve performance, and using an empty string when the teacher summary hurts performance, is a practical way to avoid teaching the compressor to produce unnecessary or harmful text.

Weakness:
I still have several concerns.

1. The paper does not do enough to explain why CORE beats full-context RAG.
The main empirical result is strong, but the paper does not really unpack where the gain is coming from. From the current experiments, it is hard to tell whether the improvement comes from removing retrieval noise, selecting only the most useful facts, rewriting the evidence into a form that is easier for the downstream model to use, or implicitly learning a prompt style that better matches the answer format. Since the paper’s core message is essentially that a shorter compressed context can work better than the full retrieved context, I think the analysis needs to go beyond final accuracy numbers. Some breakdown of what the summaries are actually doing would make the claim more convincing. For example, it would help to categorize the compressed outputs by function, or to compare against settings where only gold passages are provided, so that the role of noise reduction could be isolated more clearly.

2. The evaluation is still centered on extractive-style QA.
Although the paper includes transfer experiments and some additional results on LongBench QA tasks, the reward is ultimately defined in terms of exact match and token-level F1. That makes sense for factoid QA, but it leaves open the question of whether the same framework would work in more open-ended settings such as long-form generation, summarization, or dialogue. I do not think the paper needs to solve those settings, but it should at least discuss more explicitly how the reward would need to change, and whether the same framework is likely to remain effective outside extractive QA.

3. The baseline comparison is not tightly controlled for token budget.
The reported methods operate at noticeably different compression lengths. In Table 1, for example, CORE is around 40 tokens, LongLLMLingua is around 150 tokens, and NoiseFilter-IB is also around 40 tokens. Without comparing methods under the same token budget, it is difficult to separate the effect of better compression from the effect of simply compressing more or less aggressively. A cleaner way to make the case would be to show accuracy versus compression tradeoff curves under matched budgets.

---

> ### Author Rebuttal · Authors · 2026-03-31
>
> **Hi, Reviewer bpDj:**
>
> Thank you for acknowledging the **intuitive motivation, strong method-problem alignment, practical design, strong transferability, sensible data curation, and comprehensive experiments** of CORE.
>
> Below, we provide detailed responses to your concerns.
>
> **`Re. to W1: Analyzing Why CORE Works Better`**
>
> In fact, we have already included relevant analyses in the current manuscript. That said, we fully agree that these analyses should be further expanded in response to your excellent suggestions.
> - **Our case studies (Section 4.8; Tables 9 and 10) show what the summaries are actually doing.** They not only help explain why CORE outperforms baselines, but also demonstrate that CORE simultaneously serves several functions raised in the reviewer’s comment:
>   - **Removing retrieval noise**: In Table 9, the original retrieved documents are filled with misleading temporal distractors (e.g. 1971, 1972) and irrelevant narrative fluff. CORE successfully strips away this noise.
>   - **Selecting useful facts**: In Table 9, CORE pinpoints the exact correct year (1973) among various dates, and in Table 10, it identifies the specific grandfather among numerous family members.
>   - **Cross-document reasoning & rewriting evidence**: In Table 10, CORE performs multi-hop reasoning across separate documents, deduces the chain of evidence and rewrites it into a single, straightforward premise.
> - **Additional analysis of noise reduction (Appendix D.2)** shows CORE has denoising capability.
> - **Action for Revision**: We will add a quantitative analysis that categorizes the compressed outputs by function (e.g., noise filtering, fact selection).
>
> **`Re. to W2: Discussion on open-ended settings`**
>
> - For open-ended settings, the reward should be replaced with evaluation criteria appropriate for generative tasks. For long-form generation, rewards should focus on instruction-following and coherence, scored via an LLM-as-a-Judge or preference reward models. For summarization, the reward should balance information coverage and factual consistency using metrics like BARTScore or ROUGE, together with hallucination penalties. For dialogue, the reward can be turn-level coherence and persona consistency. Importantly, only the reward changes; the framework itself remains the same: optimizing a lightweight compressor for downstream task performance. **Therefore, CORE should generalize naturally beyond extractive QA to open-ended settings with appropriate rewards**.
>
> - ​​**Action for Revision**: We will add a subsection to explicitly detail how reward functions can be adapted for open-ended settings.
>
> **`Re. to W3: Token budget`**
>
> Our experiments used the default settings of each baseline, which may result in different compression lengths.
>
> We additionally conducted a **budget-controlled evaluation**.
>
> **Budget=50**
> ||#tok|EM|
> |-|-|-|
> |NoiseFilter|38|27.97|
> |LongLingua|46|27.12|
> |QGC|45|29.16|
> |CORE|36|33.67|
>
> **Budget=100**
> ||#tok|EM|
> |-|-|-|
> |NoiseFilter|86|28.06|
> |LongLingua|92|27.68|
> |QGC|83|29.45|
> |CORE|85|33.91|
>
> The results show: 1) CORE outperforms all baselines under the same token budget. 2) As the budget increases from 50 to 100, methods show modest gains. 3) CORE’s advantage does not come from a more favorable compression level, but from retaining more useful information under the same budget.
>
> **`Re. to Q1: Training cost and reproducibility`**
>
> - **The process of generating teacher summaries is highly efficient in both time and cost.** By utilizing a script that makes batched concurrent calls to the DeepSeek API, processing all 79000 samples in NQ dataset takes only **1.5 h**, incurs a total cost of just **$20.8**, and requires **no GPU resources**.
>
> - Appendix B reports the RL training takes 5 h on 8 H20 GPUs. The distillation SFT takes less than 0.5 h. Therefore, the **total cost is 1.5+0.5+5=7 hours**.
>
> - An anonymous GitHub link was provided in our paper for easy reproduction.
>
> **`Re. to Q2: Smaller teacher`**
>
> - Following the reviewer’s suggestion, **we additionally conducted an ablation using a smaller teacher, Qwen3-32B**. The results show that although smaller teacher leads to only a slight drop in performance, **our method still outperforms the full-context baseline**. This suggests that our method is not sensitive to the choice of teacher model. **We will add the results to the final version**.
>
> ||NQ|2Wiki|
> |-|-|-|
> |full-context|38.03|29.64|
> |CORE-(DS)|41.02|36.72|
> |CORE-(Qwen)|40.67|36.08|
>
> **`Re. to Q3: Seed variance and reward trajectories`**
>
> - We conducted **three RL runs with different random seeds** and report the EM score on NQ. The std of 0.15 indicates low variance.
>
> |Seed|CORE|
> |-|-|
> |42 (verl default)|41.02|
> |2026|41.18|
> |10|40.89|
> |Mean±Std|41.03±0.15|
>
> - **We provide reward trajectories for two training runs** at https://anonymous.4open.science/r/CORE-28B4/README.md. In both runs, the rewards increase steadily and eventually stabilize, suggesting **stable training and convergence**.

---

### Decision · Program_Chairs · 2026-04-30

**Decision:**

Accept (regular)

**Comment:**

The paper proposes core RAG, that moves away from proxy heuristics and instead uses a performance-driven learning framework. It treats context compression as an optimization problem, via a two stage training policy: 1. knowledge distillation and 2. RL. The results show significant benefit with context retention of only 3-6%, and often improving the average exact matching score by 3.3 points. The reviewers overall remained positive. Out of the reviewers who gave -ve score, R bpDj did not acknowledge the rebuttal, an the AC thinks the authors did a good job in responding to bpDj's concerns. On the other hand the major concern of sCqz on demonstrating on Wikipedia, has been decently defended. Overall the AC considers this as a submission worthy of acceptance.